# Millipede genomes reveal unique adaptations during myriapod evolution

Zhe Qu[1◉], Wenyan Nong[1◉], Wai Lok So[1◉], Tom Barton-Owen[1◉], Yiqian Li[1◉], Thomas C. N. Leung [2], Chade Li[1], Tobias Baril [3], Annette Y. P. Wong[1], Thomas Swale[4], Ting-Fung Chan [2], Alexander Hayward[3], Sai-Ming Ngai [2], Jerome H. L. Hui [1]*

1 School of Life Sciences, Simon F.S. Li Marine Science Laboratory, State Key Laboratory of Agrobiotechnology, The Chinese University of Hong Kong, Hong Kong, 2 School of Life Sciences, State Key Laboratory of Agrobiotechnology, The Chinese University of Hong Kong, Hong Kong, 3 Department of Conservation and Ecology, Penryn Campus, University of Exeter, Exeter, United Kingdom, 4 Dovetail Genomics, Scotts Valley, California, United States of America

◉ These authors contributed equally to this work.
* jeromehui@cuhk.edu.hk

**Data Availability Statement:** The final genome assemblies have been deposited to NCBI database with accession numbers JAAFCF000000000 and JAAFCE000000000. The mRNA and sRNA

## Abstract

The Myriapoda, composed of millipedes and centipedes, is a fascinating but poorly understood branch of life, including species with a highly unusual body plan and a range of unique adaptations to their environment. Here, we sequenced and assembled 2 chromosomal-level genomes of the millipedes *Helicorthomorpha holstii* (assembly size = 182 Mb; shortest scaffold/contig length needed to cover 50% of the genome [N50] = 18.11 Mb mainly on 8 pseudomolecules) and *Trigoniulus corallinus* (assembly size = 449 Mb, N50 = 26.78 Mb mainly on 17 pseudomolecules). Unique genomic features, patterns of gene regulation, and defence systems in millipedes, not observed in other arthropods, are revealed. Both repeat content and intron size are major contributors to the observed differences in millipede genome size. Tight Hox and the first loose ecdysozoan ParaHox homeobox clusters are identified, and a myriapod-specific genomic rearrangement including *Hox3* is also observed. The Argonaute (AGO) proteins for loading small RNAs are duplicated in both millipedes, but unlike in insects, an AGO duplicate has become a pseudogene. Evidence of post-transcriptional modification in small RNAs—including species-specific microRNA arm switching—providing differential gene regulation is also obtained. Millipedes possesses a unique ozadene defensive gland unlike the venomous forcipules found in centipedes. We identify sets of genes associated with the ozadene that play roles in chemical defence as well as antimicrobial activity. Macrosynteny analyses revealed highly conserved genomic blocks between the 2 millipedes and deuterostomes. Collectively, our analyses of millipede genomes reveal that a series of unique adaptations have occurred in this major lineage of arthropod diversity. The 2 high-quality millipede genomes provided here shed new light on the conserved and lineage-specific features of millipedes and centipedes. These findings demonstrate the importance of the consideration of both centipede and millipede genomes—and in particular the reconstruction of the myriapod ancestral situation—for future research to improve understanding of arthropod evolution, and animal evolutionary genomics more widely.

transcriptomic data generated in this study have been deposited to the NCBI database under the following BioProject accessions: PRJNA564202 (*Helicorthomorpha holstii*) and PRJNA564195 (*Trigoniulus corallinus*).

**Funding:** This work was supported by Hong Kong Research Grants Council (RGC) General Research Fund (GRF) (14103516, 14100919) and The Chinese University of Hong Kong (CUHK) Direct Grant (4053248). The funders had no role in study design, data collection and analysis, decision to publish, or preparation of the manuscript.

**Competing interests:** The authors have declared that no competing interests exist.

**Abbreviations:** ACN, acetonitrile; AGO, Argonaute; ANTP, Antennapedia; ARSB, quinone-less arylsulfatase b; BGI, Beijing Genomics Institute; BUSCO, Benchmarking Universal Single-Copy Ortholog; COI, cytochrome oxidase subunit; CpG, region of DNA where a cytosine nucleotide is followed by a guanine nucleotide; DRSC, *Drosophila* S2 cell; EVM, EVidenceModeler; gDNA, genomic DNA; HNL, hydroxynitrile; LINE, long interspersed nuclear element; LTR, long terminal repeat; MOX, mandelonitrile oxidase; MRP, quinone-less multidrug resistance protein; N50, shortest scaffold/contig length needed to cover 50% of the genome; PO, phenoloxidase; QO, quinone oxidase; SE, single end; SINE, short interspersed nuclear element; siRNA, small interfering RNA; TE, transposable element; tRNA, transfer RNA; UTR, untranslated region; VTG, vitellogenin-like.

## Background

Arthropoda comprises the myriapods (millipedes and centipedes), crustaceans (shrimps, crabs, and lobsters), chelicerates (spiders, scorpions, and horseshoe crabs), and insects. Collectively, these taxa account for the majority of described terrestrial and aquatic animal species on earth (Fig 1A). While crustaceans, chelicerates, and insects have been the focus of intense research, the myriapods are comparatively much less studied, despite their great diversity and important ecological roles. In particular, arthropod genomic and transcriptomic information is highly uneven, with a heavy bias toward the crustaceans, chelicerates, and insects [1–2]. Yet myriapods display many interesting biological characteristics, including a multisegmented trunk supported by an unusually large number of legs.

Centipede is Latin for "100 feet," but centipedes have between 30 and 354 legs, and no species has exactly 100 legs [5]. In contrast, millipede is Latin for "1,000 feet," and while millipedes include the "leggiest" animal on Earth, no species has as many as 1,000 legs, with the true number varying between 22 and 750 legs [6]. Myriapods were among the first arthropods to invade the land from the sea, during an independent terrestrialisation from early arachnids and insects, which occurred during the Silurian period approximately 400 million years ago [7]. Today, the Myriapoda consists of approximately 16,000 species, all of which are terrestrial [8]. Currently, just 2 myriapod genomes are available: the centipede *Strigamia maritima* [9] and a draft genome of the millipede *Trigoniulus corallinus* [10]. Consequently, the myriapods, and particularly the millipedes, present an excellent opportunity to improve understanding of arthropod evolution and genomics.

Millipedes compose the class Diplopoda, a highly diverse group containing more than 12,000 described species [11]. Millipedes are important components of terrestrial ecosystems, especially regarding their roles in the breakdown of organic plant materials and nutrient recycling. In contrast to centipedes, which have one pair of legs per body segment, individual body segments are fused in pairs in millipedes, resulting in a series of double-legged segments (diplo-segments). The typical millipede body plan consists of the head, the collum (the first posterior segment next to the head which is legless, from the Latin for "neck"), and trunk (the remaining length of the body, with varying numbers of diplo-segments) [12]. The primary

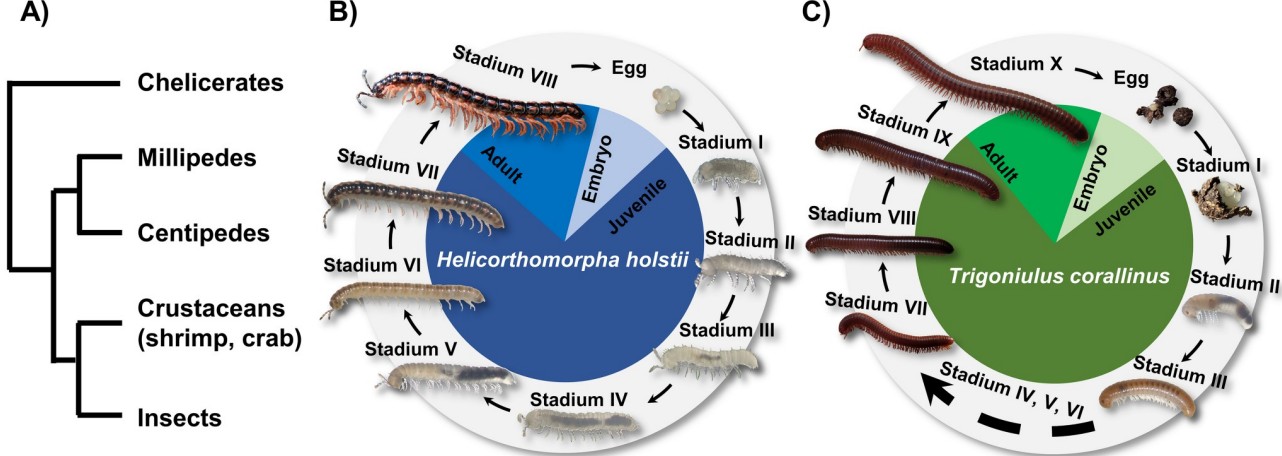

**Fig 1. Millipede phylogenetic position and life cycle.** (A) Schematic diagram showing the phylogeny of myriapods, crustaceans, and insects; (B) life cycle of the orange rosary millipede, *Helicorthomorpha holstii* (based on [3] and our observations); (C) life cycle of the rusty millipede, *Trigoniulus corallinus* (based on [4]).

defence mechanisms of millipedes are their tough exoskeletons and their ability to curl up into a tight coil, while a unique secondary defence system in some species involves emitting toxic liquids or gases from the ozadene gland, via ozopores located on each side of the posterior portion of diplo-segments termed "metazonites" [13].

The orange rosary millipede *Helicorthomorpha holstii* (Polydesmida) and the rusty millipede *T. corallinus* (Spirobolida) were chosen in this study to represent 2 major lineages from among the 16 orders of millipede. Both species originate in Asia but are now cosmopolitan species with large distributions worldwide. *H. holstii* undergoes development with a fixed number of legs and segments that increases at every stadium after each molt, and it completes 7 juvenile stadia before reaching sexual maturity at stadium VIII (adult) (Fig 1B). Conversely, *T. corallinus* undergoes development with a variable number of new segments and legs added during the initial molts, with no further segments developing after reaching stadium X (adult) (Fig 1C).

Here, we present 2 high-quality de novo reference genomes close to the chromosome-level assembly, for the orange rosary millipede *H. holstii* and the spirobolid rusty millipede *T. corallinus* (Table 1). With reference to these genomes, we reveal the basis of a unique defence system, alongside novel features of the genome and gene regulation in millipedes, that are not observed in other arthropods. The genomic resources we develop expand the known gene repertoire of myriapods and provide a genetic toolkit for furthering understanding of their unique adaptations and evolutionary pathways.

## Results and discussion

### Evolution of millipede genomic composition and size

Genomic DNA (gDNA) was extracted from single individuals of 2 species of millipedes: the orange rosary millipede *H. holstii* (Fig 1B) and the rusty millipede *T. corallinus* (Fig 1C). gDNA was sequenced using Illumina short-read and 10X Genomics linked-read sequencing platforms (S1 Table). Hi-C libraries were also constructed for both species and sequenced on the Illumina platform (S1 Fig). Both genomes were first assembled using short reads, followed by scaffolding with Hi-C data. The *H. holstii* genome assembly is 182 Mb with a shortest scaffold/contig length needed to cover 50% of the genome (N50) of 18.11 Mb (Table 1, S2 Table). This high physical contiguity is matched by a high completeness, with a 97.2% complete Benchmarking Universal Single-Copy Ortholog (BUSCO) score for eukaryotic genes

**Table 1. Comparison of myriapod genome assembly quality.**

| Common name | Coastal centipede | Orange rosary millipede | Rusty millipede | Rusty millipede |
|---|---|---|---|---|
| Species name | *Strigamia maritima* | *Helicorthomorpha holstii* | *Trigoniulus corallinus* | *Trigoniulus corallinus* |
| Accession number | GCA_000239455.1 | JAAFCF000000000 | JAAFCE000000000 | PRJNA260872 |
| Assembly size | 176,210,797 | 181,201,347 | 448,558,750 | 416,979,918 |
| Scaffold N50 | 139,451 | 18,119,263 | 26,787,286 | N/A |
| Number of scaffolds | 14,739 | 7,137 | 9,127 | N/A |
| Contig N50 | 24,745 | 335,075 | 184,856 | 955 |
| Number of contigs | 24,080 | 16,022 | 27,543 | 1,233,936 |
| Gap content (N) | 1.48% | 1.95% | 1.42% | 0.4% |
| Number of genes | 15,461 | 23,013 | 21,361 | N/A |
| Complete BUSCOs | 96.7% | 97.7% | 97.2% | 46.2% |
| Reference | [9] | This study | This study | [10] |

**Abbreviations:** BUSCO, Benchmarking Universal Single-Copy Ortholog; N50, shortest scaffold/contig length needed to cover 50% of the genome; N/A, Not available

(Table 1). The *T. corallinus* genome is 449 Mb with a scaffold N50 of 26.7 Mb and 96.7% BUSCO completeness (Table 1, S2 Table). There were 23,013 and 21,361 gene models predicted for the *H. holstii* and *T. corallinus* genome assemblies, respectively (Table 1). The majority of sequences assembled for the *H. holstii* and *T. corallinus* genomes are contained on 8 and 17 pseudomolecules, respectively (S1 Fig). A total of 4 out of 7 pseudomolecules for *H. holstii* and 11 out of 17 pseudomolecules for *T. corallinus* contain telomeric repeats (S1 Data), illustrating that these genome assemblies represent the first close to chromosomal-level genomes available for myriapods.

Transposable elements (TEs) are almost ubiquitous components of eukaryotic genomes, often accounting for a large proportion of an organism's genome [14]. Among metazoans, arthropods are a particular focus for TE research. However, myriapods are the only major branch of Arthropoda for which knowledge of TEs remains extremely poor. Here, we examined the repeat content of 1 centipede and 2 millipede genomes to perform the first comparative investigation of TEs in the Myriapoda. As for other major arthropod groups [15], we find considerable variation in the total genomic contribution and composition of TEs among myriapod genomes. TE content accounts for 19% to 47% of the total assembled genome among the 3 available myriapod genomes, with total repeat content (including more simple repeat categories) varying between 19% and 55% ("Repeat Content" of Fig 2; S1 Text, S3 Table). In the millipede *T. corallinus*, repeats account for more than half of all gDNA, representing 55% (245 Mb) of the total assembled genome size (S2 Data). This finding is of interest since the genome of *T. corallinus* is more than double the size of either of the other 2 myriapod genomes at 449 Mb, compared with 182 Mb in *H. holstii* and 176 Mb in *S. maritima* ("Repeat Content" of Fig 2). In contrast, repeat content is 40% (70 Mb) in the centipede *S. maritima* and just 19% (35 Mb) in the millipede *H. holstii*, demonstrating considerable variation among myriapod lineages (S2 Data). TE expansions, particularly of long interspersed nuclear elements (LINEs) and DNA elements, appear to have played a role in genome size expansion in *T. corallinus*. However, given that genome size remains approximately 28% greater in *T. corallinus* after exclusion of all annotated repeats, it is evident that additional explanations must underlie the greater genome size for *T. corallinus* (i.e., *T. corallinus*: 449 Mb genome − 245 Mb repeat content = 204 Mb; *H. holstii*: 182 Mb genome − 35 Mb repeat content = 147 Mb). We find that variation in genome size is further accounted for by a considerably greater total for coding regions in *T. corallinus* (sum of coding regions = 212 Mb), due primarily to an expansion in intron size, compared to *H. holstii* (sum of coding regions = 78 Mb) (S4 Table). Thus, we find that both repeat content and intron size are major contributors to the observed differences in millipede genome size.

## Homeobox gene organisation rearrangement

The most celebrated feature of millipedes is their many body segments and accompanying extreme number of legs. Homeobox genes are an ideal candidate to study body plan evolution, including segment number, as they are conserved gene expression regulators in animals. To understand whether the 3 myriapod lineages considered here share a similar number of homeobox genes, we compared their homeobox gene content to that of all available insect genomes. We found that the 3 myriapod genomes have undergone 3 lineage-specific duplications of common homeobox genes (*Otx*, *Barhl*, *Irx*) (S2 Fig). These data suggest that millipede genomes have not undergone massive homeobox gene duplications comparable to that which have occurred in the centipede genome.

Hox gene clusters are renowned for their role in the developmental patterning of the anteroposterior axis of animals. In both *H. holstii* and *T. corallinus* genomes, intact Hox clusters

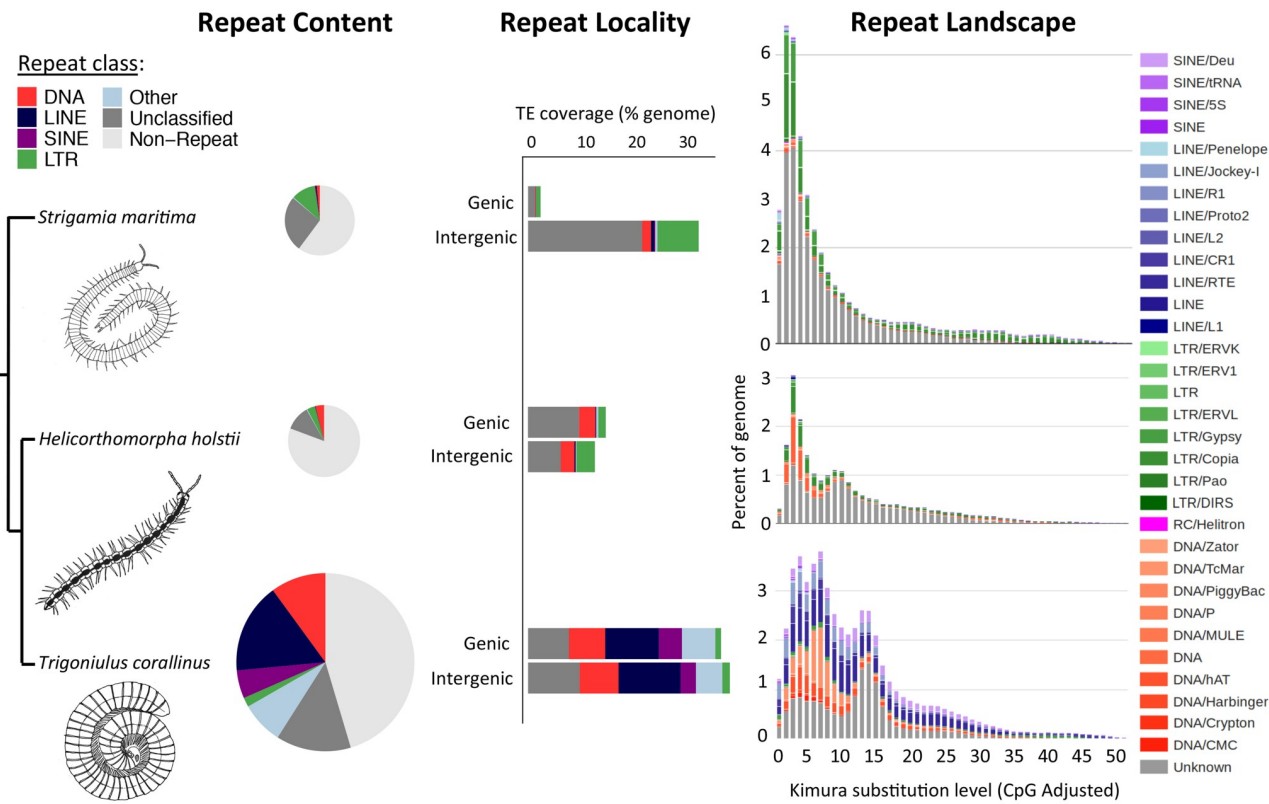

**Fig 2. TE content, genomic locality, and estimates of accumulation history for sequenced members of the Myriapoda.** Phylogenetic relationships among taxa are indicated on the left-hand side of the figure, alongside schematics of each myriapod species. From left to right: (i) pie charts scaled in proportion to assembled genome size, illustrating the relative contribution to myriapod genomes from each major repeat class; (ii) stacked bar charts illustrating the proportion of each repeat class found in genic (≤2 kb from an annotated gene) versus intergenic regions (>2 kb from an annotated gene) for each myriapod species, expressed as a percentage of the total assembled genome; (iii) repeat landscape plots illustrating TE accumulation history for each myriapod genome, based on Kimura distance-based copy divergence analyses, with sequence divergence (CpG adjusted Kimura substitution level) illustrated on the x-axis, percentage of the genome represented by each TE type on the y-axis, and transposon type indicated by the colour chart on the right-hand side. The underlying data of this figure can be found in S8 Data. CpG, region of DNA where a cytosine nucleotide is followed by a guanine nucleotide; LINE, long interspersed nuclear element; LTR, long terminal repeat; SINE, short interspersed nuclear element; TE, transposable element; tRNA, transfer RNA.

containing orthologues of most arthropod Hox genes were recovered (except for the *Hox3* gene), and we show that these are expressed during early developmental stages (Fig 3, S3 and S4 Figs). In the *H. holstii* genome, no *Hox3* orthologues could be identified, and 2 *Hox3* genes were located together on a different scaffold to the Hox cluster scaffold in *T. corallinus*. This situation mirrors that observed previously in the genome of the centipede *S. maritima* [9] (Fig 3).

Segmentation and tagmosis (the formation of tagmata through fusion and modification of several individual segments) are considered to be key drivers for the evolutionary success of arthropod adaptive radiations [16]. Changes in Hox gene evolution are linked to these processes [17]. In particular, *Hox3* has been an important player in arthropod evolution. For example, *Hox3* has undergone tandem duplication to form the 3 copies *bicoid*, *zen*, and *z2* in dipterans, and it has duplicated extensively in lepidopterans to take up novel roles [18]. Unlike the fast evolving *Hox3* genes documented in these insects, the homeodomain sequences of *Hox3* duplicates in both the centipede *S. maritima* and the millipede *T. corallinus* are conserved [19] (S5 Fig). In both the centipede *Lithobius atkinsoni* and the millipede *Glomeris*

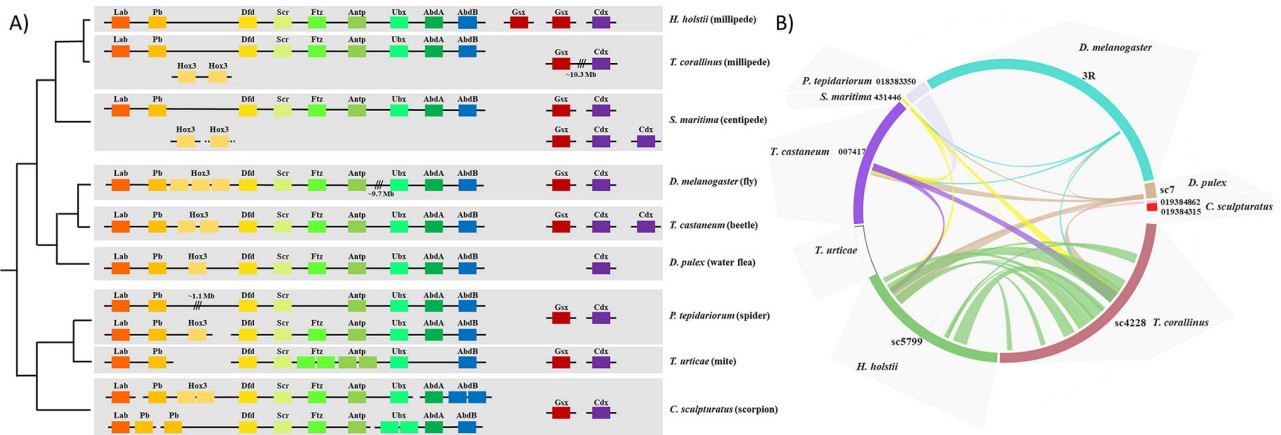

**Fig 3. Homeobox gene clusters.** (A) Hox and ParaHox gene cluster genomic organisations in millipedes and other arthropods. (B) Synteny comparisons between Hox gene scaffolds. Mindots = Minimum number of genes required to define a syntenic block.

*marginata*, *Hox3* appears to have a typical Hox-like role [20,21]. In our transcriptomic analyses, both the *Hox3A* and *Hox3B* genes were found to be expressed at higher levels early in development compared to latter developmental stages, while *Hox3A* is expressed in the egg stage, and *Hox3B* is expressed in both the egg and stadium II (S4 Fig). Future investigations on the spatiotemporal expression of *Hox3* genes in myriapod embryos will be helpful to examine further whether they have typical Hox-like roles. Since *H. holstii* contains no *Hox3* orthologues while both *T. corallinus* and *S. maritima* have 2 *Hox3* genes outside their Hox clusters, we conclude that the genomic "relaxation" of *Hox3* from intact tight Hox clusters appears to have occurred in the ancestor of all myriapods (Fig 3).

The ParaHox cluster is the paralogous sister of the Hox cluster and contains an array of 3 Antennapedia (ANTP)-class homeobox genes: *Gsx/ind*, *Xlox/Pdx*, and *Cdx/cad*. Together, these genes are responsible for patterning the brain and endoderm formation in bilaterians [22]. In general, the genomic linkage of ParaHox cluster genes has been lost in all investigated ecdysozoans [23]. However, we identify a loosely linked ParaHox cluster of *Gsx* and *Cdx* in the millipede *T. corallinus*, representing the first identified ecdysozoan ParaHox cluster (Fig 3). ParaHox genes are expressed mainly during early development, and *Gsx* is also expressed during a late developmental stage (S4 Fig). Given that ParaHox clustering has been identified in the lophotrochozoans and deuterostomes [22,23], our data provide evidence that arthropod and ecdysozoan ancestors contained clustering of ParaHox genes, rather than having disintegrated ParaHox clusters as previously thought.

Similar to the situation in *S. maritima*, the Eve orthologue is closely linked to the Hox clusters in both millipede genomes. In addition, in the millipede genomes examined here we identified the linkage of other ANTP-class homeobox gene members to Hox-Eve, including the genes *Abox*, *Exex*, *Dll*, *Nedx*, *En*, *Unpg*, *Ro*, and *Btn* (S6 Fig). Other homeobox gene clusters were also identified and compared, including the NK cluster and the Irx cluster (S6 Fig). Whether this situation represents a difference in genome quality between the 2 millipede genomes (N50 = 26.7 Mb and 18.1 Mb) and the centipede genome [9] (N50 = 139 kb) or a true difference in genomic content between millipede and centipede lineages remains to be tested following improvements to centipede genomic resources.

In both millipedes, homeobox genes are generally expressed early in development rather than during later developmental stages. Nevertheless, homeobox genes in *H. holstii* generally

have higher expression levels in stadium II than in the egg stage, while the situation is reversed in *T. corallinus* (S4 Fig). Whether these differences in observed expression patterns represent distinctions in the developmental modes between polydemids and spirobolids remains to be tested by comparison to additional millipede genomes.

Collectively, the earlier examples of homeobox gene organisation highlight the importance of the novel genomic resources presented here to (1) reconstruct the arthropod ancestral situation, by providing novel interpretations of lineage-specific modifications, and (2) understand functional constraints acting in extant lineages, such as the relaxation in *Hox3* and *Xlox* in the ecdysozoan Hox and ParaHox clusters. As additional high-quality genomic resources become available for a wider sampling of myriapod lineages, it will become possible to test the patterns identified here more generally.

## Conserved and divergent microRNA regulation and machinery

To understand how post-transcriptional regulators have evolved in myriapods, small RNA transcriptomes were obtained from eggs, juveniles, and adults of *H. holstii* and *T. corallinus* (S5 Table). Using stringent criteria to annotate microRNAs supported by small RNA reads, a total of 59 and 58 conserved microRNAs were identified in the genomes of *H. holstii* and *T. corallinus*, respectively (S4–S6 Data). This number is comparable to the 58 microRNAs identified in the centipede *S. maritima* [9]. In addition to conserved microRNAs, 43 and 10 novel lineage-specific microRNAs could further be identified in millipedes *H. holstii* and *T. corallinus*, respectively, with only one conserved in both species (S8 Fig, S4, S5 and S6 Data). No homologue of miR-125, a member of the ancient bilaterian miR-100/let-7/miR-125 cluster, could be identified in the centipede *S. maritima* [9, 24]. However, miR-125 could be identified in both millipede genomes, suggesting a lineage-specific loss in the centipede (Fig 4A). In addition, our 2 high-quality millipede genomes allowed us to reveal the presence of conserved microRNA clusters, including miR-100-let-7-125, miR-263-96, miR-283-12, miR-275-305, miR-317-277-34, miR-71-13-2, miR-750-1175, and miR-993-10-iab4/8, as observed in most arthropods (S4 Data). Previously, miR-283 has been identified in pancrustaceans only, but it could be identified in the 2 millipede genomes presented here (S4 Data). Moreover, miR-96 and miR-2001 could be identified in the 2 millipedes, but not in *S. maritima* (S4 Data). These examples highlight the importance of having multiple high-quality myriapod genomes to properly understand the comparative evolution of post-transcriptional regulators, which will ultimately allow us to address fundamental questions regarding the evolution of microRNA

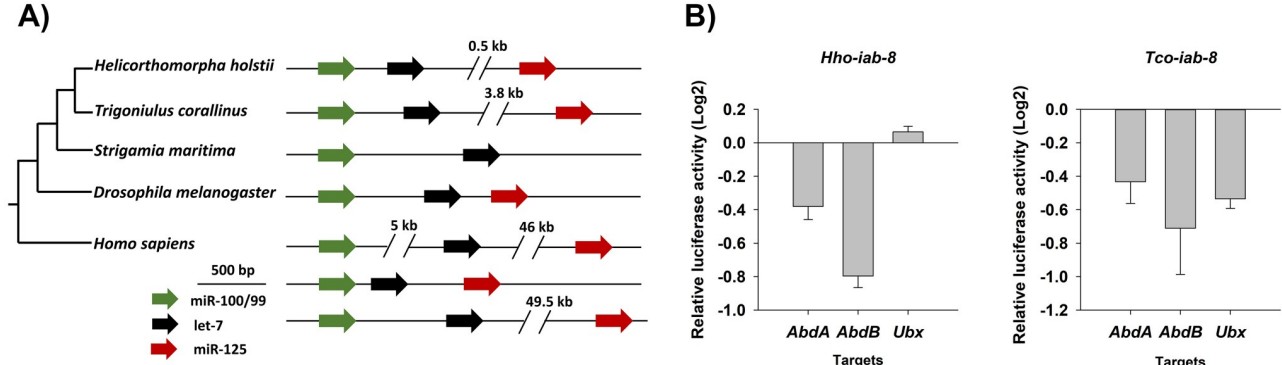

**Fig 4. MicroRNAs in millipedes.** (A) Genomic organisation of miR-100/let-7/miR-125 clusters in various animals; (B) luciferase assays showing the repression activities of Hox genes by miR-iab-8 in both millipedes. The underlying data of this figure can be found in S8 Data.

regulation and associated machinery. In *Drosophila melanogaster*, the iab-4/iab-8 locus encodes bi-directionally transcribed microRNAs that regulate the function of flanking Hox transcription factors. We show that bi-directional transcription, temporal and spatial expression patterns and Hox regulatory function of the iab-4/iab-8 locus are conserved between fly and the beetle *Tribolium castaneum*. Computational predictions suggest iab-4 and iab-8 microRNAs can target common sites, and cell-culture assays confirm that iab-4 and iab-8 function overlaps on Hox target sites in both fly and beetle. However, we observe key differences in the way Hox genes are targeted. For instance, abd-A transcripts are targeted only by iab-8 in *Drosophila*, whereas both iab-4 and iab-8 bind to *Tribolium* abd-A. Our evolutionary and functional characterization of a bi-directionally transcribed microRNA establishes the iab-4/iab-8 system as a model for understanding how multiple products from sense and antisense microRNAs target common sites.

Comparing conserved microRNAs between the 2 millipedes considered here, and all available insect genomes with small RNA sequencing data, we found multiple cases of microRNAs undergoing microRNA arm switching (S2 Text, S9 Fig and S4 Data). Two cases including iab-8 and miR-2788 are further analysed in detail in S2 Text, S10 and S11 Figs. Given the swapping of arm usage and the separate genes targeted by the different arms, governance of microRNA arm switching could operate under multiple mechanisms and present an additional and potentially underappreciated means of adaptation.

We further explored how conserved microRNAs may modulate gene regulatory networks among arthropod lineages. In insects, the bidirectionally transcribed microRNA iab-4/iab-8 locus is renowned for regulating the functions of its flanking Hox genes in the genomic cluster [25]. In both the millipedes *H. holstii* and *T. corallinus*, the microRNA iab-4/iab-8 locus is located between Hox genes *abd-A* and *abd-B*, similar to the situation in insects. Using a cell-based dual-luciferase reporter assay to test Hox gene targets targeted by iab-8 in the 2 millipedes, we found that the posterior Hox genes can be down-regulated in the 2 millipedes (abd-A and abd-B by *H. holstii* iab-8, abd-A, abd-B and Ubx by *T. corallinus* iab-8) as found in insects (Fig 4B). These data further establish that the regulation of Hox genes *Ubx* and *abd-A* is partially performed by the microRNA iab-8 in the most recent common ancestor of insects and myriapods. The gain and loss of transcription of Hox genes, such as alternation of *Ubx* expression has been linked to the evolutionary transition to the hexapod limb pattern [26]. Whether the evolution of iab4/8 Hox targets as demonstrated here could be a plausible mechanism to generate the diversity of morphology of myriapods is an exciting question that remains to be further tested.

Another informative candidate to improve understanding of animal evolution are the small RNAs and their associated machineries. Small RNAs represent an additional set of conserved gene expression regulators in animals, and their study can reveal hidden layers of gene regulation. For example, mature microRNAs are 21–23 nucleotide noncoding RNAs that regulate gene expression and translation, usually binding onto the 3′ untranslated regions (UTRs) of target mRNAs to achieve post-transcriptional inhibition, either by suppressing translation or inducing mRNA degradation [27, 28] (Fig 5A). Despite the finding that the biogenesis pathways of microRNAs and other small RNAs are relatively conserved in animals, modifications of small RNA machinery has been found to alter small RNA regulation and thus contribute to the rewiring of genetic networks; for example, the placozoan *Trichoplax adhaerens* has lost the *Piwi*, *Pasha*, and *Hen1* genes, and no microRNAs are known to be produced [29].

All genes responsible for small RNA machinery were identified in the 2 millipede genomes generated here, along with an unusual duplication of the Argonaute (*Ago*) gene, while the other biogenesis components (Dgcr8, Drosha, Exportin-5, TRBP) remain the same (Fig 5A, S12, S13 and S14 Figs). In insects, it is well known that there are also 2 *Ago* forms, and in the

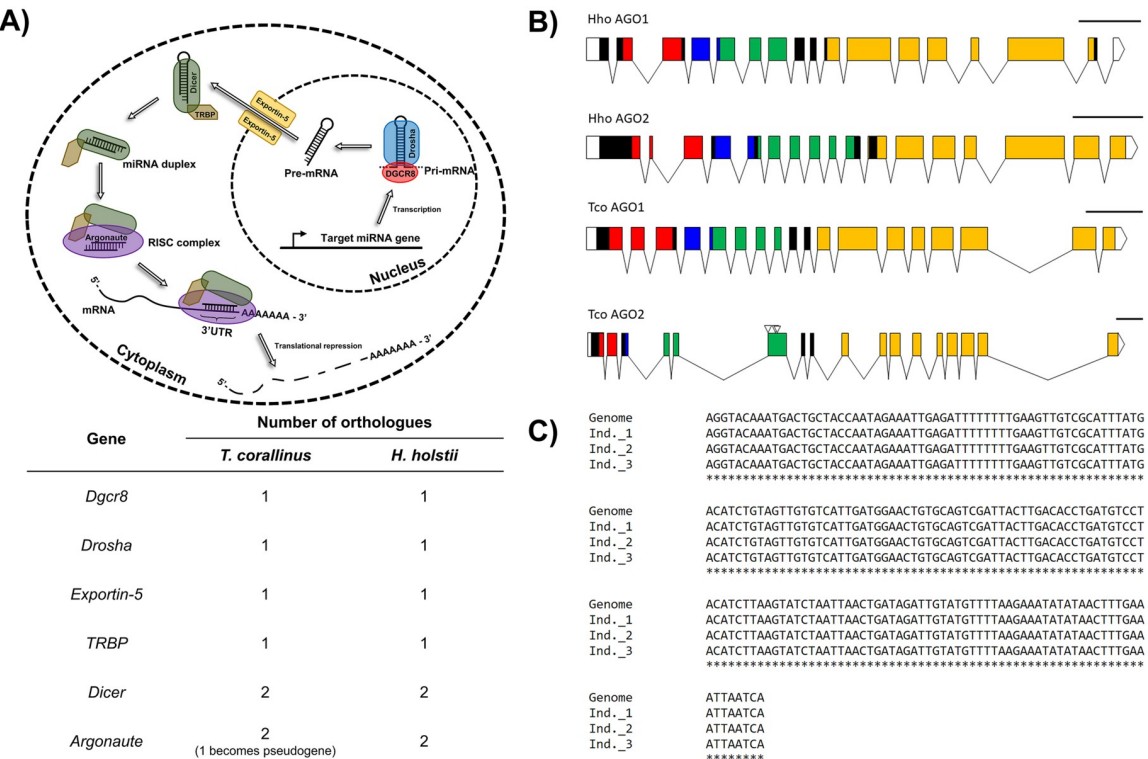

**Fig 5. Argonaute duplication in millipedes.** (A) Schematic diagram showing the biogenesis pathway of microRNAs (upper) and a table summarising the number of gene copies contained in each millipede genome (lower); (B) schematic diagram showing the duplicates of the Ago gene in the 2 genomes. Conserved domains of AGO—ArgoN (red), ArgoL (blue), PAZ (green), and PIWI (orange). Inverted triangles, in TcoAGO2, indicate the position of multiple stop codons found in the corresponding gene sequence. Scale bar = 500 nucleotides. (C) Confirmation of the TcoAGO2 pseudogene. PCR and Sanger sequencing were carried out on gDNA collected from 3 *Trigoniulus corallinus* individuals that were not used for genome sequencing. Ago, Argonaute; gDNA, genomic DNA; miRNA, microRNA; UTR, untranslated region.

fruit fly *Drosophila melanogaster*, the dominant arm of the precursor microRNA can be sorted into *Ago1* to direct translational repression. Meanwhile, the other microRNA arm, as well as small-interfering RNA (siRNA), can be sorted into *Ago2* to direct transcriptional degradation [30–34]. However, phylogenetic analyses suggest that the Ago duplication in millipedes is lineage specific and does not share the same origin as the duplication event that occurred in insects (S12 Fig). In addition, one Ago in *T. corallinus* (which we named *Ago2*) appears to have become a pseudogene, as a consequence of a sequence insertion resulting in multiple stop codons (Fig 5B). To test that this finding was not due to a genome sequencing error, or mutation in a single individual, we carried out PCR and Sanger sequencing on 3 additional individuals and confirmed this insertion mutation across all individuals (Fig 5C). The Ago copy in *H. holstii* retains an open reading frame and so is likely to be functional, but it is currently unclear what the evolutionary significance of the duplication event is.

## The ozadene: Millipede chemical defence

Many millipedes possess ozadene glands, which are specialised secretory integumental sacs arranged segmentally along the body. Ozadenes synthesize, store, and exude a diverse cocktail of chemicals, including alkaloids, quinones, phenols, and cyanogenic compounds [35, 36]. Ozadenes are considered primarily to be an anti-predator adaptation, since their secretions

include paralyzing agents, repellents, toxins, and sticky substances that provide physical protection, but they have also been suggested to be provide a defence against pathogenic microbes [35]. However, little fine-scale analysis of ozadene genetics has been undertaken. Consequently, we used the millipede genomic resources generated here to further investigate the functional genomics of the ozadene.

Ozadene glands are divided into 3 main types based on their morphology and the chemicals that they produce: glomerid-, julid-, and polydesmid-type ozadenes [35]. The polydesmid-type ozadene, such as that found in *H. holstii*, is a complex structure consisting of a large membranous sac, a narrow valved duct leading to a reaction chamber, and an ozopore that opens onto the body surface (certain segments only from the 5th to 19th segment) [35, 37]. Many polydesmid millipedes are known to secrete cyanogenic compounds such as hydrogen cyanide as a chemical defence [36]. Genes involved in the biosynthetic pathway of cyanide—including *CYP3201B1*, mandelonitrile oxidase (*MOX*), α-hydroxynitrile (*HNL*), and β-glucosidase, as well as rhodanese, which is involved in its detoxification—were identified in the *H. holstii* genome (Fig 6A, S7 Data). This represents one of the most complete sets of cyanogenic pathway genes identified in a single millipede species to date. Surprisingly, despite an absence of reports demonstrating the capability of polydesmid millipedes to synthesize quinones as defensive chemicals, quinone orthologues were also identified in the genome of *H. holstii* (Fig 6B, S7 Data). However, due to the size of the gland as well as the availability of animals, mass spectrometry was not carried out on the gland of *H. holstii*.

Julid-type ozadene glands of the type found in *T. corallinus* are relatively simple structures in comparison to polydesmid-type ozadenes [12, 35] (Fig 6C). Spirobolid millipedes, including *T. corallinus*, are well known to secrete quinones, including benzoquinone and hydroquinone, as chemical defence [35,36]. Orthologues of phenoloxidase (*PO*), quinone oxidase (*QO*), vitellogenin-like (*VTG*), quinone-less arylsulfatase b (*ARSB*), and quinone-less multidrug resistance protein (*MRP*) were identified in the *T. corallinus* genome (Fig 6B, S7 Data). Strikingly, a total of 2,125 peptides were identified by mass spectrometry in the *T. corallinus* ozadene gland, with VTG peptide being the most abundant (S7 Data). The function of this VTG peptide is unknown. Further, a total of 119 proteins were identified to possess antibacterial, antifungal, or antiviral properties (S15 Fig), corroborating results from other studies suggesting that millipede ozadenes also play an important antimicrobial role [42–44]. Consequently, our data add weight to the assertion that one of the main functions of the millipede ozadene, at least in *T. corallinus*, is to provide defence against pathogenic microorganisms, in addition to providing a defence mechanism against predators.

## Conserved synteny among myriapod and deuterostome genomes

Conservation of large-scale gene linkage has been reported between the centipede *S. maritima* and the amphioxus *Branchiostoma floridae* at a higher level than with any insect, providing evidence that the last common ancestor of arthropods retained significant synteny with the last common ancestor of bilaterians [9]. To understand the genomic rearrangement patterns among diplopods and chilopods, we performed conserved synteny analyses between the 2 millipede genomes sequenced here and the centipede *S. maritima*. As expected, a greater level of conserved synteny blocks were detected between the 2 millipedes than between the millipede and the centipede, while greater large-scale gene linkage was observed between *T. corallinus* and *S. maritima* than between *H. holstii* and *S. maritima* (S16 Fig). To specifically test for the presence of conserved syntenic blocks between millipede genomes and deuterostomes, as observed for the centipede genome *S. maritima*, we also compared syntenic relationships between the 3 myriapod genomes to those of a variety of deuterostome genomes, including

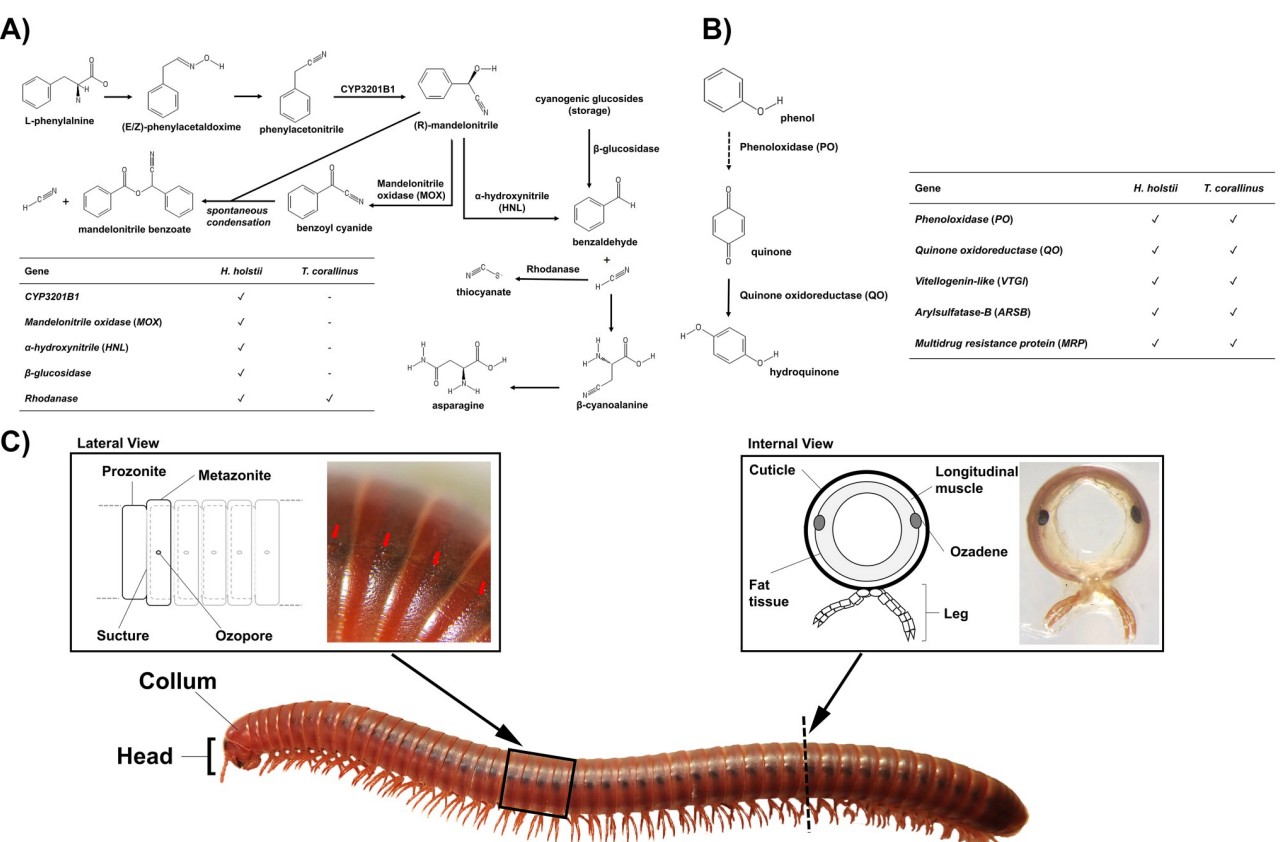

**Fig 6. Chemical defence in millipedes.** Schematic diagram of the metabolic pathways of (A) HCN and benzaldehyde and (B) quinone in millipedes. The pathways are drawn based on previous studies [38–41]. (C) The ozadene defensive gland of the millipede *T. corallinus*. HCN, hydrogen cyanide.

human, amphioxus, tunicate, and sea urchin (S17, S18, S19 and S20 Figs). As shown in Fig 7, syntenic blocks could be detected between both millipede genomes and deuterostome genomes. These data highlight the importance of millipede genomes as a reference for the reconstruction of animal evolutionary history.

## Conclusions

The 2 chromosomal-level millipede genomes provided in this study considerably expand our genomic understanding of myriapods, a diverse and ecologically important invertebrate group with a key phylogenetic position among the arthropods. Compared to the single available genome for their closest relatives—the centipedes—our findings highlight that millipede genomes have retained distinct ancestral features (e.g., synteny to human) and have undergone unique changes in their genomic machinery (e.g., *Hox3*, *Xlox*, AGO proteins, microRNAs). We also show that millipede genomes display considerable variability in their repeat content and genome size, and we begin to unravel the genomic bases of some of their morphological adaptations. Future research focussing on understanding arthropod evolution—and, in particular, the reconstruction of the myriapod ancestral situation for comparison to other clades—will require further genomic resources for both centipedes and millipedes.

In *Drosophila melanogaster*, the iab-4/iab-8 locus encodes bi-directionally transcribed microRNAs that regulate the function of flanking Hox transcription factors. We show that bi-

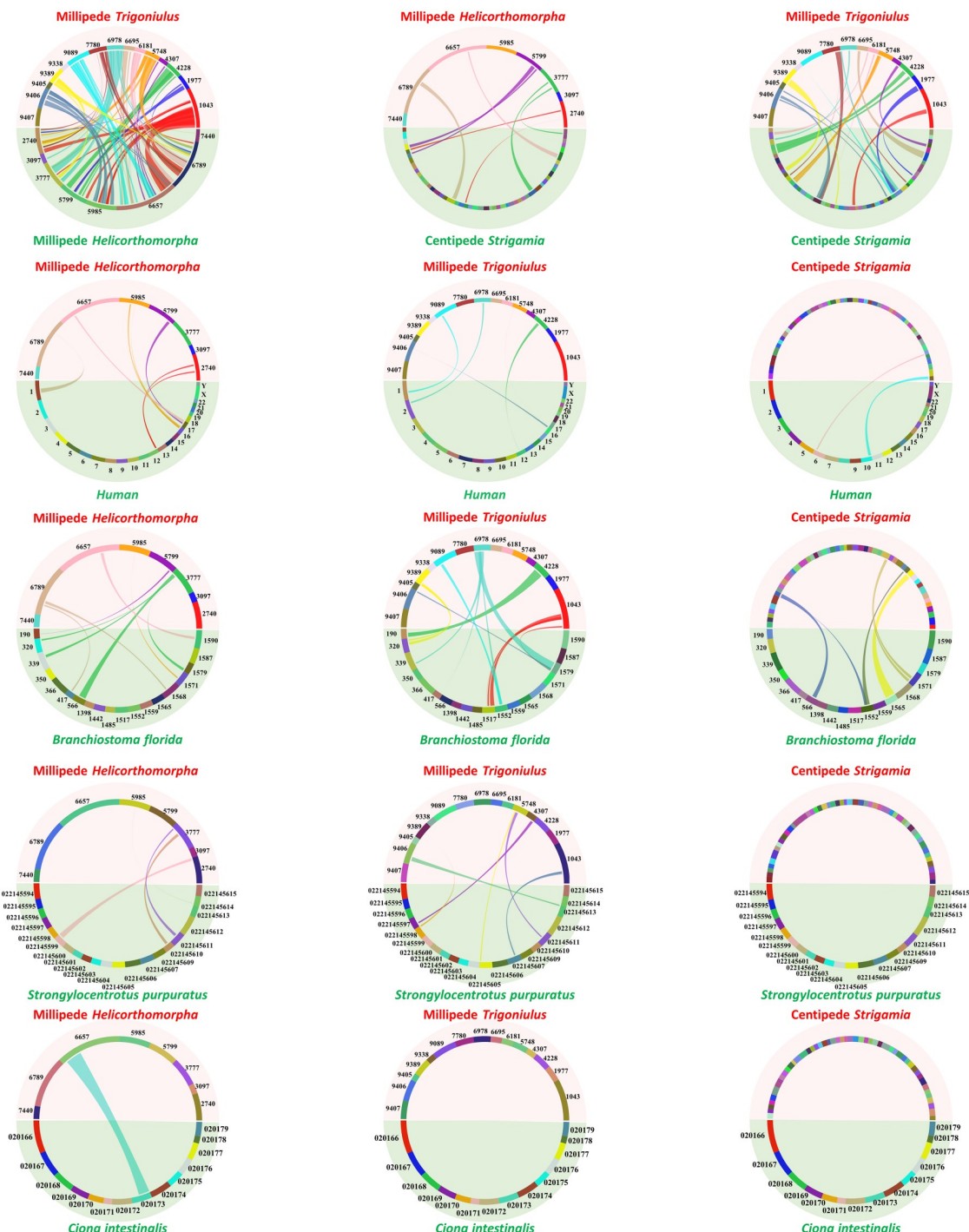

**Fig 7. Synteny comparisons of myriapod (millipedes *Helicorthomorpha* and *Trigoniulus* and centipede *Strigamia*) and deuterostome genomes.** Note that different degrees of syntenic regions could be detected between millipede genomes and deuterostome genomes, and between the centipede genome to deuterostome genomes.

directional transcription, temporal and spatial expression patterns and Hox regulatory function of the iab-4/iab-8 locus are conserved between fly and the beetle *Tribolium castaneum*. Computational predictions suggest iab-4 and iab-8 microRNAs can target common sites, and cell-culture assays confirm that iab-4 and iab-8 function overlaps on Hox target sites in both fly and beetle. However, we observe key differences in the way Hox genes are targeted. For instance, abd-A transcripts are targeted only by iab-8 in *Drosophila*, whereas both iab-4 and iab-8 bind to *Tribolium* abd-A. Our evolutionary and functional characterization of a bi-directionally transcribed microRNA establishes the iab-4/iab-8 system as a model for understanding how multiple products from sense and antisense microRNAs target common sites.

In *Drosophila melanogaster*, the iab-4/iab-8 locus encodes bi-directionally transcribed microRNAs that regulate the function of flanking Hox transcription factors. We show that bi-directional transcription, temporal and spatial expression patterns and Hox regulatory function of the iab-4/iab-8 locus are conserved between fly and the beetle *Tribolium castaneum*. Computational predictions suggest iab-4 and iab-8 microRNAs can target common sites, and cell-culture assays confirm that iab-4 and iab-8 function overlaps on Hox target sites in both fly and beetle. However, we observe key differences in the way Hox genes are targeted. For instance, abd-A transcripts are targeted only by iab-8 in *Drosophila*, whereas both iab-4 and iab-8 bind to *Tribolium* abd-A. Our evolutionary and functional characterization of a bi-directionally transcribed microRNA establishes the iab-4/iab-8 system as a model for understanding how multiple products from sense and antisense microRNAs target common sites.

## Materials and methods

### Animal husbandry

Adult *T. corallinus* were captured locally in an agricultural garden of New Asia College of The Chinese University of Hong Kong. Species identity was confirmed by DNA Sanger sequencing of the mitochondrial cytochrome oxidase subunit I (COI) gene, with a pair of universal primers, LCO1490 and HCO2198 [45]. Millipedes were collected under the soft and humid soil of decaying organic matter in a grass field and kept in a 39 cm (width) × 52 cm (length) × 27 cm (depth) plastic aquarium at room temperature. The aquarium was filled with slightly moistened gardening soil as the base substrate. Hydrated dried sphagnum moss was provided as a source of water. Dried leaves collected together with the millipedes were first boiled and then transferred on top of the soil. Apple slices were provided occasionally with a small quantity of Zoo Med's Repti Calcium powder. Distilled water was sprayed in a 2-day interval to maintain soil humidity.

Adult *H. holstii* were collected locally in a garden of Sin Hang Ho College of The Chinese University of Hong Kong. Species identity was confirmed by DNA Sanger sequencing of the mitochondrial COI gene, as described earlier for *T. corallinus*. Millipedes were captured on the surface of tree trunks or the rocky substrate and were subsequently kept in a 40 cm (width) × 57 cm (length) × 15 cm (depth) plastic aquarium at room temperature. The provision of substrate and food source was the same as described for *T. corallinus*. In addition, several autoclaved rocks were placed in the aquarium to mimic the environment of their original collection sites. Unlike *T. corallinus*, *H. holstii* requires a drier environment, therefore distilled water was sprayed at a 4-day interval to provide adequate moisture.

### Genome sequencing

gDNA was extracted from male *T. corallinus* and male *H. holstii* excluding the digestive tract, using a PureLink Genomic DNA Mini Kit (Invitrogen) following the manufacturer's protocol. Extracted gDNA was subjected to quality control using gel electrophoresis. Qualifying samples

were sent to Novogene and to Dovetail Genomics for library preparation and sequencing. In addition, a Chicago library was prepared by Dovetail Genomics using the method described by Putnam and colleagues [46]. Briefly, approximately 500 ng of high–molecular-weight gDNA (mean fragment length = 55 kb) was reconstituted into artificial chromatin in vitro and fixed with formaldehyde. Fixed chromatin was digested with DpnII, the 5′ overhangs were filled in with biotinylated nucleotides, and then free blunt ends were ligated. After ligation, crosslinks were reversed, and the DNA was purified from the protein. Purified DNA was treated to remove biotin that was not internal to ligated fragments. The DNA was then sheared to approximately 350 bp mean fragment size, and sequencing libraries were generated using NEBNext Ultra enzymes and Illumina-compatible adapters. Biotin-containing fragments were isolated using streptavidin beads before PCR enrichment of each library. Libraries were sequenced on an Illumina HiSeq X platform. Dovetail HiC libraries were prepared in a similar manner as described previously [47]. Briefly, for each library, chromatin was fixed with form-aldehyde in the nucleus, and then extracted fixed chromatin was digested with DpnII, 5′ over-hangs were filled in with biotinylated nucleotides, and free blunt ends were ligated. After ligation, crosslinks were reversed and the DNA purified from protein. Purified DNA was treated to remove biotin that was not internal to ligated fragments. The DNA was then sheared to approximately 350 bp mean fragment size, and sequencing libraries were generated using NEBNext Ultra enzymes and Illumina-compatible adapters. Biotin-containing fragments were isolated using streptavidin beads before PCR enrichment of each library. Details of the sequencing data can be found in S1 Table.

### Transcriptome sequencing

Transcriptomes of multiple developmental stages of each species were sequenced at Novogene. Total RNA from different tissues was isolated using TRIzol reagent (Invitrogen) according to the manufacturer's instructions and quality controlled using a Nanodrop spectrophotometer (Thermo Scientific), gel electrophoresis, and Agilent 2100 Bioanalyzer (Agilent RNA 6000 Nano Kit). Qualifying samples underwent library construction and sequencing at Novogene; polyA-selected RNA sequencing libraries were prepared using TruSeq RNA Sample Prep Kit version 2. Insert sizes and library concentrations of final libraries were determined using an Agilent 2100 Bioanalyzer instrument (Agilent DNA 1000 Reagents) and real-time quantitative PCR (TaqMan Probe), respectively. Small RNA (<200 nt) was isolated using the mirVana miRNA isolation kit (Ambion) according to the manufacturer's instructions. Small RNA was dissolved in the elution buffer provided in the mirVana miRNA isolation kit (Thermo Fisher Scientific) and submitted to Novogene for HiSeq Small RNA library construction and 50 bp single-end (SE) sequencing. Details of the sequencing data can be found in S5 Table.

### Sequencing data pre-processing

For Illumina sequencing data, adapters were trimmed, and reads were filtered using the following parameters with custom scripts carried out by the sequencing company: "-n 0.1" (if N accounted for 10% or more of reads) and "-l 4 -q 0.5" (if the quality value is lower than 4 and accounts for 50% or more of reads). FastQC was run for quality control [48]. If adapter contamination was identified, adapter sequences were removed using Minion [49]. Adapter trimming and quality trimming was then performed with Cutadapt version 1.10 [50].

### Estimation of genome characteristics

For each species, k-mers of the Illumina PE library of 200 bp and 170 bp insert size of *H. holstii* and *T. corallinus* were counted using Jellyfish version 2.2.5 with k-mers = 31 [51], and

estimation of genome size, repeat content, and heterozygosity were analysed based on a kmer-based statistical approach using GenomeScope [52]. Kraken was used to estimate the percentage of reads originating from bacterial contamination [53], and 1.45% and 0.52% sequencing reads were marked as bacteria and removed from *H. holstii* and *T. corallinus*, respectively.

### *H. holstii* and *T. corallinus* genome assembly

Chromium WGS reads were separately used to make a de novo assembly using Supernova (version 2.1.1), with the command "—maxreads = 231545066" for *T. corallinus* and "—maxreads = 100000000" for *H. holstii*. The de novo assembly, shotgun reads, Chicago library reads, and Dovetail HiC library reads were used as input data for HiRise, a software pipeline designed for using proximity ligation data to scaffold genome assemblies [46]. An iterative analysis was conducted. First, Shotgun and Chicago library sequences were aligned to the draft input assembly using a modified SNAP read mapper (http://snap.cs.berkeley.edu). The separation of Chicago read pairs mapped within draft scaffolds was analysed by HiRise to produce a likelihood model for genomic distance between read pairs, and the model was used to identify and break putative misjoins, to score prospective joins, and to make joins above a threshold. After aligning and scaffolding Chicago data, Dovetail HiC library sequences were aligned and scaffolded following the same method. After scaffolding, shotgun sequences were used to close gaps between contigs.

### Gene model prediction

Raw sequencing reads of the transcriptomes were pre-processed with Trimmomatic (version 0.33; with parameters "ILLUMINACLIP:TruSeq3-PE.fa:2:30:10 SLIDINGWINDOW:4:5 LEADING:5 TRAILING:5 MINLEN:25") [54]. The genome sequences were first cleaned and masked by Funannotate [55]. The soft masked assembly was then aligned to RNA sequencing data using Trinity [56] and PASA [57]. The PASA gene models were then used to train Augustus in "funannotate predict" step following manufacturer-recommended options for eukaryotic genomes (https://funannotate.readthedocs.io/en/latest/tutorials.html#non-fungal-genomes-higher-eukaryotes). Several prediction sources—including GeneMark [58], high-quality Augustus predictions (HiQ), PASA [57], Augustus [59], GlimmerHMM [60], and snap [61]—were then passed to EVidenceModeler (EVM) [57] with EVM Weights {'GeneMark': 1, 'HiQ': 2, 'pasa': 6, 'proteins': 1, 'Augustus': 1, 'GlimmerHMM': 1, 'snap': 1, 'transcripts': 1}, and the final annotation files were generated.

### TE annotation

Repetitive elements were identified using an in-house pipeline. Firstly, elements were identified using RepeatMasker version 4.0.8 [62] with the *Arthropoda* RepBase [63] repeat library. Low-complexity repeats were ignored (-nolow), and a sensitive (-s) search was performed. Following this, a de novo repeat library was constructed using RepeatModeler version 1.0.11 [62], including RECON version 1.08 [64] and RepeatScout version 1.0.5 [65]. Novel repeats identified by RepeatModeler were analysed with a "BLAST, Extract, Extend" process to characterise elements along their entire length [66]. Consensus sequences and classification information for each repeat family were generated. The resulting de novo repeat library was utilised to identify repetitive elements using RepeatMasker. Repetitive element association with genomic features was determined using BedTools version 2.26.0 [67]. "Genic" repetitive elements were defined as those overlapping loci annotated as genes ± 2 kb and identified using the BedTools window function. All plots were generated using Rstudio version 1.2.1335 [8] with R version 3.5.1 [68] and ggplot2 version 3.2.1 [69].

## Gene family analysis

Gene family sequences were first retrieved from the 2 millipede genomes using tBLASTn [70]. The identity of each putatively retrieved gene was then tested by comparison to sequences in the NCBI nr database using BLASTx. For homeobox genes, sequences were retrieved with BLASTp using homeodomain sequences retrieved from HomeoDB [71]. For phylogenetic analyses of gene families, DNA sequences were translated into amino acid sequences and aligned to other members of the respective gene family, gapped sites were removed from alignments using MEGA, and phylogenetic trees (maximum likelihood, maximum parsimony, and neighbor joining) were constructed using MEGA [72] and IQTree [73].

## Synteny analyses

Synteny blocks between genomes of the 2 millipedes, centipede *S. maritima* [9], human *Homo sapiens* (NCBI Assembly GCF_000001405.38), amphioxus *B. floridae* [74], sea urchin *Strongylocentrotus purpuratus* (NCBI_Assembly GCA_000002235.4), and tunicate *Ciona intestinalis* [75] were computed using SyMAP version 4.2 (Synteny Mapping and Analysis Program) [76], with the parameter "mask_all_but_genes" set to 1 to mask the non-genic sequences and "Mindots" (the minimum number of anchors required to define a synteny block) set between 2 and 7.

## Ozadene mass spectrometry and analyses

Ozadene glands were isolated from adult *T. corallinus* under a dissecting stereomicroscope in 1X phosphate-buffered saline (Gibco, Life Technologies) with forceps, and protein samples were dissolved in sample buffer (7 M urea, 2 M thiourea, 0.1 M DTT) and alkylated with 5 mM iodoacetamide for 30 minutes in the dark at room temperature given the light- and temperature-sensitive reagents. Following this, sequencing-grade trypsin (Promega) was added to each sample at a 1:20 ratio and incubated overnight at 37 ˚C. The digests were then mixed with the same volume of SCX buffer (20 mM $KH_2PO_4$/50% acetonitrile [ACN; pH 3.0]) and loaded onto an SCX spin column (Thermo Fisher Scientific). Peptides were eluted with 1M KCl in 10 mM $KH_2PO_4$/25% ACN (pH 3.0) and dried in a SpeedVac. Then the SCX cleaned-up samples were re-suspended in 0.1% trifluoroacetic acid and fractionated into 4 fractions with increasing ACN concentrations (7.5%, 12.5%, 17.5%, 50%) using a high-pH reversed-phase fractionation kit (Thermo Fisher Scientific). The nano-LC separation was performed using a Dionex UltiMate 3000 RSLC nano system. Following this, 1 μg of peptide was loaded onto a 25-cm-long, 75-μm-internal-diameter C18 column and eluted at a constant flow rate of 0.3 μL/min with a linear gradient from 2% to 35% of ACN over 2 hours. Eluted peptides were analysed using an Orbitrap Fusion Lumos Tribrid mass spectrometer (Thermo Fisher Scientific). Survey scans (MS) and data-dependent scans (MS/MS) were acquired in the Orbitrap Fusion Lumos Tribrid Mass Spectrometer (Thermo Fisher Scientific) with a mass resolution of 60,000 and 15,000, respectively. MS scan range was from 375 to 1,500 m/z. The AGC targets for MS and MS/MS were 4e5 and 5e4, respectively; the maximum injection times for MS and MS/MS were 50 ms and 250 ms, respectively. Precursor isolation windows were set to 1.6 m/z. Data were analysed by Proteome Discoverer version 2.4 with SEQUEST as a search engine. The searching parameters were as follows: oxidation of methionine (+15.9949 Da) and carbamidomethylation of cysteine (+57.0215 Da) was set as dynamic modification; precursor-ion mass tolerance, 10 ppm; fragments-ion mass tolerance, 0.02 Da. Proteins were quantified utilizing the precursor ion quantification module of Proteome Discoverer. The Enzyme Commission Numbers (EC Number) of the proteins were assigned using EggNOG-Mapper 1.0.3 with Egg-NOG 5.0; protein families were classified with InterProScan 5.40–77.0.

## Small RNA analyses

Adaptor sequences were trimmed from small RNA sequencing reads, and Phred quality scores less than 20 were removed. Processed reads of length 18 bp to 27 bp were then mapped to respective genomes using the mapper.pl module of the mirDeep2 package [77]. To identify known microRNAs, predicted millipede microRNA hairpins were compared against metazoan microRNA precursor sequences from miRBase [78] using BLASTn (e-value < 0.01) [70]. Those microRNAs with no significant sequence similarity to any of the microRNAs in miR-Base were checked manually. Novel microRNAs were defined when they fulfilled the criteria of microRNAs [79] (MirGeneDB 2.0, https://mirgenedb.org/information).

The expression levels of different arms of a microRNA were calculated based on the number of sequencing reads mapped to the respective arm region in the predicted microRNA hairpin by bowtie. To compare arm usage among various insects and millipedes, all published small RNA sequencing data sets for insects with a genome available were used. miRNA hairpins from both miRBase [78] and InsectBase [80] were used for read mapping and counting. The micro-RNAs with either arm with absolute counts >50 were included in the arm switching analysis. The formula $\omega = 5p \div (5p + 3p)$, in which 5p and 3p refer to the number of predicted 5p arms and 3p arms, respectively, was adopted as the measure of arm selection value. The arm selection value, $\omega$, ranged from 0 to 1, with smaller values indicating the tendency of 3p preference and larger values indicating the tendency of 5p preference. We adopted <0.3 as the value of 3p dominance and >0.7 as the value of 5p dominance [24, 25]. To define the arm preference of a micro-RNA from a species with multiple sRNA sequencing samples, the arm dominance was decided by majority of the dominance observed in all samples (more than 70% of total samples).

To validate the preference of microRNA 5p and 3p arms, candidate microRNA hairpins with 100–300 bp sequence flanks were amplified from respective gDNA and cloned into a pAC5.1 vector (Invitrogen) (primer information is shown in S6 Table). Also, the binding site (perfect complementary to mature microRNA) of each arm was cloned into a psicheck-2 vector (Promega) (primer information is shown in S6 Table). All constructs were sent to Beijing Genomics Institute (BGI)–Hong Kong for sequencing for confirmation of their identities before use. *Drosophila* S2 cells (DRSCs) were kept in Schneider *Drosophila* medium (Life Technologies) with 10% (v/v) heat-inactivated foetal bovine serum (Gibco, Life Technologies) and 1:100 penicillin-streptomycin (Gibco, Life Technologies) at 23 ˚C. The psicheck-2 vector with binding sites of each microRNA arm (500 ng) and pAC5.1-microRNA (100 ng) were co-transfected into DRSCs using Effectene (Qiagen). Forty-eight hours after transfection, lucifer-ase activities were measured using the Dual-Luciferase Reporter Assay System (Promega) and a Tecan Infinite M200 luminometer. The *Renilla* firefly luciferase activity ratios were calcu-lated and normalized to control that DRSCs were transfected with the respective psicheck-2-binding site alone. Subsequently, the relative luciferase activity of the 5p binding site to the 3p binding site was calculated. Three biological replicates were carried out for each test.

For microRNA-target validation, the 3′ UTRs of predicted target genes were amplified and subcloned into a psicheck-2 vector (primer information is shown in S6 Table). Cell transfec-tion and dual-luciferase reporter assays were carried out as described previously in DRSCs using the psicheck-2-3′UTR and pAC5.1-microRNA [25, 81]. The *Renilla* firefly luciferase activity ratios were calculated and normalized to control that DRSCs were transfected with the respective psicheck-2-3′UTR alone.

## Supporting information

**S1 Fig. Hi-C information of two millipede genome assemblies.** Hi-C information of *H. hol-stii* (a) and *T. corallinus* (b). The x- and y-axes give the mapping positions of the first and

second read in the read pair, respectively, grouped into bins. The colour of each square gives the number of read pairs within that bin. White vertical and black horizontal lines have been added to show the borders between scaffolds. Scaffolds less than 1 Mb are excluded.
(PDF)

**S2 Fig. Homeobox gene information.** Specific gains of homeobox genes between myriapods and insects (a) and between centipede and millipedes (b). A total of 108 and 105 homeobox genes could be identified in the genomes of *H. holstii* and *T. corallinus*, respectively, which is comparable to the 112 homeobox genes that could be identified in the centipede *S. maritima* [9].
(PDF)

**S3 Fig. Syntenic analyses of Hox gene scaffolds in various arthropods.**
(PDF)

**S4 Fig. Heatmap showing expression level of homeobox genes during *H. holstii* and *T. corallinus* development.**
(PDF)

**S5 Fig. Amino acid alignment of the bilaterian Hox3 and Xlox genes.**
(PDF)

**S6 Fig. Homeobox genes arrangement.** (a) Schematic diagram showing the ANTP-class homeobox gene arrangement in the 2 millipede genomes; (b) schematic diagram showing the homeobox gene clusters in the myriapod genomes. (c) Schematic diagram showing the homeobox genes in the 2 millipede genomes. Details of the genomic locations and sequences of all these homeobox genes can be found in S3 Data. Gene tree can be found in S7 Fig. Hh, *H. holstii*; Sm, *S. maritima*; Tc, *T. corallinus*.
(PDF)

**S7 Fig. Phylogenetic tree of Homeobox genes.** The sequence alignment can be found in S8 Data.
(PDF)

**S8 Fig. Conserved sequences between the novel microRNA identified in *H. holistii* and *T. corallinus*.**
(PDF)

**S9 Fig. MicroRNA arm switching.** Cases of microRNAs undergone arm switching in insect and millipede genomes (a). Red boxes represent 5p arm dominance, blue boxes represent 3p arm dominance, yellow boxes represent cases for which microRNA dominant arm cannot be determined based on the cut-off set up in this study, and grey boxes represent multiple copies of microRNAs in respective genomes and so cases of arms usage are not determined; (b) microRNA arm switching cases of let-7 (left) and miR-277 (right) in insects. The red and blue colour represent the 5p arm and 3p arm, respectively. Data underlying this figure can be found in S8 Data.
(PDF)

**S10 Fig. iab-8 microRNA arm switching.** (a) Differential arm target repression ability by different arthropod species of miR-iab-8. Bars represent mean with SEM; *t* test was used to determine significant difference between 5p and 3p. $^*p < 0.05$; (b–d) predicted number of targets and their GO of miR-iab-8-5p and miR-iab-8-3p in fly and millipedes. The underlying data of this figure can be found in S8 Data. Dme, *D. melanogaster*; GO, gene ontology; Hho, *H. holistii*; Tco, *T. corallinus*.
(PDF)

**S11 Fig. miR-2788 microRNA arm switching.** (a–b) Small RNA read counts of miR-2788 in different developmental stages in millipede *H. holstii* and in TcA cell line of beetle *Tribolium castaneum*; (c–d) luciferase activity showing the differential arm target (i.e., miR-2788-5p and -3p sensor) repression ability between miR-2788 carrying different flanking sequence of *H. holstii* and *T. castaneum*. Bars represent mean with SEM; (e–f) predicted number of targets and their GO of miR-2788-5p and miR-2788-3p in beetle *T. castaneum* and millipede *H. holstii*. The underlying data of this figure can be found in S8 Data. FA, adult female; GO, gene ontology; J17, juvenile; MA, adult male; S1–S7, stadia I–VII; TcA, TcA cell line.
(PDF)

**S12 Fig. Phylogenetic tree of AGO proteins.** Alignments of protein sequences were made with MUSCLE and the tree built with MEGA 7.0, with 1,000 bootstrap replicates. Maximum likelihood (black), maximum parsimony (red), and neighbour joining (blue) algorithm were adopted, and corresponding high-confidence bootstrap values are shown. The sequence alignment can be found in S8 Data. AGO, Argonaute.
(PDF)

**S13 Fig. Phylogenetic tree of DGCR8.** Alignments of protein sequences were made with MUSCLE and the tree built with MEGA 7.0, with 1,000 bootstrap replicates. Maximum likelihood (black), maximum parsimony (red) and neighbour joining (blue) algorithm were adopted, and corresponding high-confidence bootstrap values are shown. The sequence alignment can be found in S8 Data.
(PDF)

**S14 Fig. Phylogenetic tree of Dicer and Drosha.** Alignments of protein sequences were made with MUSCLE and the tree built with MEGA 7.0, with 1,000 bootstrap replicates. Maximum likelihood (black), maximum parsimony (red) and neighbour joining (blue) algorithm were adopted, and corresponding high-confidence bootstrap values are shown. The sequence alignment can be found in S8 Data.
(PDF)

**S15 Fig. Antimicrobial properties of peptides in ozadene of *T. corallinus*.**
(PDF)

**S16 Fig. Syntenic blocks between millipede and centipede genomes.**
(PDF)

**S17 Fig. Syntenic blocks between myriapod and human genomes.**
(PDF)

**S18 Fig. Syntenic blocks between myriapod and amphioxus *B. floridae* genomes.**
(PDF)

**S19 Fig. Syntenic blocks between myriapod and sea urchin *S. purpuratus* genomes.**
(PDF)

**S20 Fig. Syntenic blocks between myriapod and sea squirt *C. intestinalis* genomes.**
(PDF)

**S1 Table. Genome sequencing data information.**
(DOCX)

**S2 Table. Genome size prediction by GenomeScope.**
(DOCX)

**S3 Table. Comparison of TEs in the 3 myriapod genomes.**
(DOCX)

**S4 Table. Comparison of the gene sizes in the 3 myriapod genomes.**
(DOCX)

**S5 Table. Transcriptome sequencing data information of *H. holstii* and *T. corallinus*.**
(DOCX)

**S6 Table. Sequence information of primers used in this study.**
(DOCX)

**S1 Text. TEs.** TE, transposable element.
(DOCX)

**S2 Text. MicroRNAs.**
(DOCX)

**S1 Data. Telomeric repeats in the 2 millipede genomes.**
(XLS)

**S2 Data. TEs in 3 myriapod genomes.** TE, transposable element.
(XLSX)

**S3 Data. Homeobox gene sequences annotated in the 2 millipede genomes and their expression levels.**
(XLSX)

**S4 Data. The microRNA contents, arm usage, and predicted gene targets.**
(XLSX)

**S5 Data. *H. holstii* microRNA structures.**
(PDF)

**S6 Data. *T. corallinus* microRNA structures.**
(PDF)

**S7 Data. Chemical defence involving genes in millipede genomes and list of proteins identified in the *T. corallinus* ozadene gland.**
(XLSX)

**S8 Data. Numerical data underlying Figs 2 and 4B, S9b, S10a and S11a–S11d Figs, and sequence alignment for phylogenies of S7, S12, S13 and S14 Figs.**
(XLSX)

## Author Contributions

**Conceptualization:** Jerome H. L. Hui.

**Data curation:** Wenyan Nong, Tom Barton-Owen, Thomas Swale.

**Formal analysis:** Zhe Qu, Wenyan Nong, Wai Lok So, Tom Barton-Owen, Yiqian Li, Thomas C. N. Leung, Chade Li, Tobias Baril, Annette Y. P. Wong, Alexander Hayward, Sai-Ming Ngai, Jerome H. L. Hui.

**Funding acquisition:** Ting-Fung Chan, Sai-Ming Ngai, Jerome H. L. Hui.

**Investigation:** Zhe Qu, Wenyan Nong, Wai Lok So, Tom Barton-Owen, Yiqian Li, Thomas C. N. Leung, Chade Li, Tobias Baril, Annette Y. P. Wong, Thomas Swale, Ting-Fung Chan, Alexander Hayward, Sai-Ming Ngai, Jerome H. L. Hui.

**Methodology:** Zhe Qu, Wenyan Nong, Wai Lok So, Tom Barton-Owen, Yiqian Li, Thomas C. N. Leung, Chade Li, Tobias Baril, Annette Y. P. Wong, Thomas Swale, Alexander Hayward, Sai-Ming Ngai.

**Project administration:** Jerome H. L. Hui.

**Resources:** Jerome H. L. Hui.

**Supervision:** Jerome H. L. Hui.

**Validation:** Zhe Qu.

**Visualization:** Wenyan Nong, Wai Lok So, Tom Barton-Owen, Yiqian Li.

**Writing – original draft:** Alexander Hayward, Jerome H. L. Hui.

**Writing – review & editing:** Zhe Qu, Wenyan Nong, Wai Lok So, Tom Barton-Owen, Yiqian Li, Thomas C. N. Leung, Chade Li, Tobias Baril, Annette Y. P. Wong, Thomas Swale, Ting-Fung Chan, Alexander Hayward, Sai-Ming Ngai, Jerome H. L. Hui.

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
