## [Editor Report · Decision Letter 0]

3 Jan 2020

Dear Dr Hui, 

Thank you for submitting your manuscript entitled "Millipede genomes reveal unique adaptation of genes and microRNAs during myriapod evolution" for consideration as a Research Article by PLOS Biology.

Your manuscript has now been evaluated by the PLOS Biology editorial staff, as well as by an academic editor with relevant expertise, and I'm writing to let you know that we would like to send your submission out for external peer review. Please accept my apologies for the delay incurred over the holiday period.

Please re-submit your manuscript within two working days, i.e. by Jan 07 2020 11:59PM.

***Please be aware that, due to the voluntary nature of our reviewers and academic editors, manuscripts may be subject to delays due to their limited availability during the holiday season. Please also note that the journal office will be closed entirely 21st- 29th December inclusive, and 1st January 2020. Thank you for your patience.***

Kind regards,

Roli Roberts

Senior Editor

PLOS Biology

---

## [Decision Letter · Decision Letter 1]

7 Feb 2020

Dear Dr Hui,

Thank you very much for submitting your manuscript "Millipede genomes reveal unique adaptation of genes and microRNAs during myriapod evolution" for consideration as a Research Article at PLOS Biology. Your manuscript has been evaluated by the PLOS Biology editors, an Academic Editor with relevant expertise, and by two independent reviewers. We had recruited a third reviewer, but they have not been able to submit their comments in a timely fashion.

IMPORTANT: You'll see that both reviewers are broadly positive about your study, but each makes a number of requests, largely for significant improvement of the presentation of the paper. Reviewer #1 has seen your paper elsewhere, and while noting that some issues have been addressed, s/he feels that it still falls short of its potential. We *strongly* encourage you to use this further opportunity to thoroughly address these issues (the Academic Editor wonders whether it might help to move some of the secondary stories to the supplementary material and thereby enhance the main narrative). You will also see that reviewer #2 makes a large number of very constructive and specific recommendations regarding the structure and balance of the manuscript, many of which may help to remedy some of the problems identified by reviewer #1. Reviewer #2 also points out that most of your methods are in the Supplementary Information; this is not appropriate, and they should be moved to the main manuscript (there are no page constraints in PLOS Biology).

In light of the reviews (below), we will not be able to accept the current version of the manuscript, but we would welcome re-submission of a much-revised version that takes into account the reviewers' comments. We cannot make any decision about publication until we have seen the revised manuscript and your response to the reviewers' comments. Your revised manuscript is also likely to be sent for further evaluation by the reviewers.

We expect to receive your revised manuscript within 2 months. 

**IMPORTANT - SUBMITTING YOUR REVISION**

*Re-submission Checklist*

*Published Peer Review*

*PLOS Data Policy*

*Blot and Gel Data Policy*

Sincerely,

Roli Roberts

Senior Editor

PLOS Biology

REVIEWERS' COMMENTS:

Reviewer #1:

Qu et al. report on the sequencing of two species of millipedes and on a genomic analysis of these species from a number of aspects. I have previously reviewed this manuscript for a different journal. I am pleased to see that the authors have taken many of my comments from the previous review into account in this revised submission. However, several of my criticisms remain. In what follows, I recap my previous review, with relevant changes and omissions, taking into account what the authors have changed in the interim.

Myriapods in general, and specifically millipedes are significantly under-represented in the world of arthropod genomics, and two new sequenced species are a very welcome addition, with significant potential for novel evolutionary insights. The quality of the sequencing, in terms of completeness and contig size, is very high, suggesting great potential. I therefore approached the manuscript with much enthusiasm, with the expectation that this would be a manuscript worth publishing. In that first reading, I was disappointed, but my recommendation to the authors was to make the effort to improve the manuscript and to deal with its problems so as to bring it up to a level that is suitable for publication. The quality of the data the authors have and the level of analyses they have carried out shows that there is definitely the potential for an excellent paper.

The previous version of the article had many technical problems, most of which have been dealt with. However, my main impression with the manuscript was that it is unfocused. There is no clear structure and the main message is not presented clearly (to the extent that there is a main message). The authors have since added a Conclusions paragraph that puts the work into a better context, but some work is needed within the rest of the manuscript to provide a stronger focus.

What tends to happen in most genome projects is that there are lot of "little stories" that come out of the genome, and putting them together to tell a coherent "big story" (whether evolutionary or applied) is tricky. In this case, I think the authors do not provide a correct balance of stories. Some get bogged down in details (the miRNA story) whereas others are drawn in such broad brush strokes that there is no message (the ozadene story). The "big story" - what we can learn about arthropod evolution form these two new genomes - is almost completely missing.

Specific comments:

Scientific issues

1) Millipedes are said to be the third largest arthropod class after insects and arachnids (line 81-82). However, these taxa are each at a different taxonomic level, so the comparison is meaningless. The reference for this claim is a taxonomic survey of Amazonian arthropods; hardly the best reference for such a claim.

2) The section on synteny added a comparison to the human genome, following a suggestion I made to the previous version. This now tells a much more interesting story, but the story still feels under-developed. The relevant figure (Fig. 2) needs a more detailed figure legend to explain exactly what is presented in it.

3) In the beginning of the Synteny section (l. 150-151) the authors say that myriapods are interesting as the outgroup to Insecta. This is incorrect. The outgroup to Insecta are crustaceans. Myriapods are the outgroup to Pancrustacea. In the following section, myriapods are not compared to insects at all, raising the question why this is mentioned. In the next section (homeobox clusters - l. 179) the comparison to insects comes up again, but it is not clear what the context is. Why does the duplication of three homeobox genes suggest suitability for a comparison with insects?

4) I am not enough of an expert on transposable elements or on miRNA to comment on the science of these two sections. However, both sections are overly detailed and overly technical for a paper with a general focus. I got lost in many of the paragraphs and felt they were lacking context or justification. Specifically, I found the discussion of host genome evolution and TEs confusing to the non-specialist.

5) Figure 1B,C - What is the source for the life history illustrations (or the data in them, if they are original)?

6) Figure 1F - What are we seeing here? This Venn-diagram is not explained at all.

Technical issues

7) Several places throughout the text: Strigamia maritima is abbreviated as S. strigamia. This was fixed in a few places, but some instances remain.

8) There is an inconsistency in the level of plain-text vs. jargon. Some terms are explained, whereas others, which are not widely understood, are left with no explanation. An obvious example is in the second paragraph of the Introduction: Collum, prozonite, metazonite should be explained.

9) The common names of one the two species is problematic. The first species, H. holstii is referred to as "The polydesmid miilipede" as if that were its common name. This is actually a very general term, since Polydesmidae is the family, so all they are saying is that it's a member of the family. From what I could see, there is no common English name for this species, and it is usually mentioned as an Asian polydesmid. Does this species have a common name in Cantonese? Rather than leaving the species orphaned and nameless, I suggest the authors give it a name - ideally a translation of an existing common name, and that will become the name it is known by (at least in the genome literature). Rusty millipede for T. corallinus is fine.

10) Materials and Methods are understandingly brief, with most of the details being in the Supplementary sections. Nonetheless, there are some methods that should be introduced in brief since their results are discussed at length in the main text (e.g. double-luciferase assays, proteomics)

11) The references are poorly formatted using inconsistent style and are missing all italicization in gene names and species names.

Reviewer #2:

The manuscript by Qu et al. describes the sequencing, assembly, and analysis of two millipede genomes. Myriapods are underrepresented among published genomes, so this is an important contribution to the community. I would like to congratulte the authors for such a high quality assembly, with scaffold N50 around or over 20Mbp. The contig N50 remains moderate (184~335kbp), but scaffolding with Hi-C and restriction-map seemed to have played very well. 

Analyses accompanying each of the specific analyses are presented with substantial additional experiments and data. On the other hand, throughout the analyses and throughout the manuscript, the key research question (or questions for specific analyses) for studying the millipedes was unclear. I detail my comments below to clarify. 

1. Ozadene gland analysis

 Search for protein toxins and antimicrobial peptides seem odd, because ozopores of dilopods are basically well known for their chemical defence, mainly using hydrogen cyanide and benzaldehyde as the primary chemical. No specific mention of these chemical defense is given, and the authors seem to complete ignore the biosynthetic pathways for these chemical. Please provide the reasons that the analyzed species do not employ such chemical defense in their ozopores. 

 The protome analyses lack the information on the abundance of composite proteins. Since there were over 2600 peptides identified, it is not useful to treat each peptides equally. If protein-based defence is the key in these species, key proteins must be highly abundant. Please discuss the significance of each candidates based on their abundances. It is also possible that hypothetical proteins with high abundance may be the key protein.

 The search for toxins and antimicrobial defence probably contains errors. For example, Tco_003099-T1 is noted as "AB AF AV Antibiotics", but blasting this sequence against RefSeq shows that this is actually mitochondrial succinate dehydrogenase (8e-148). 

2. Conserved synteny

 Synteny blocks shown in Fig.2 is probably too strict. Dot plot analysis of Chipman et al. between B. floridae and S. maritima was able to show the distant relationships, but this figure is not being able to show such synteny even between Helicorthomorpha and Strigamia. Moreover, it is almost no use to compare synteny with the human genome. The genome size is too different to visualize in this form, and is also way too distant. What was the aim of comparing with humans and not B. floridae? Even if it is compared against B. floridae, it does not provide useful information that will greatly enhance what is already reported by Chipman et al. (because the distance between previously reported S. maritima and the two millipeds of this work is negligibly small compared to that between myriapods and chordates). Fig. 2 is very uninformative and could be in supplementary figures. 

3. Homeobox

I expected this section to include the key biological findings of this manuscript, because a substantial part (about 50%) of the Introduction is dedicated to the different morphology of centipedes and millipedes, as well as different development stages between the two millipedes analyzed. However, no such discussion is given here. Please at least discuss something about the difference in developmental stages that are found from the genome, somewhere in the manuscript, or consider removing such mention in dedicated Fig. 1, because it confuses the reader about the key research question of this paper.

In terms of the contribution of ANTP cluster Hox genes in centipede development, excellent work by Hughes and Kaufman (Development 2002) should not be ignored. How does the Hox cluster pattern or their expression link to the expansion of body segmentation in millipedes? Hox3 is functioning in the head segment in centipede, and appendages around the mouth is a morphologically variable feature in myriapods. Does the Hox3 conservation pattern at least partly link to such morphology in the studied species?

Again, I think some discussion on segmentation pattern is critical here. 

4. Transposable elements

Genome size difference of the two millipedes is striking (449MB and 182MB), but the authors try to explain this based on the expansion of TE. However, even after considering the extent of repeats (55% or 245Mb in T. corallinus and 19% or 35Mb in H. holstii), remaining 204Mb and 147Mb is still 25% of difference. Moreover, gene count is greater in H. holstii (Table1, 23013 in H. holstii, 21361 in T. corallinus. This by the way does not match Table 1.3.1, which adds to 15285 in T. corallinus and 15767 in H. holstii). Was there not a gene duplication or loss pattern (as seen in Hox3) among the two species? Number and size of intron also typically also contribute to the differences in the genome size. 

5. Small RNA analysis

Many reporter assays are reported here, but please make it clear that Drosophila culture cells are used for this assay. It is good that the authors confim the mode of action of several miRNAs, but what the authors aimed to find through these experiments were difficult to follow. If genomes are analyzed in neglected groups, they would naturally have conserved aspects and also unconserved parts. Yes, millipedes do have different usage of miRNAs compared to Insecta, but what biological aspects of millipedes do these different modes actually contribute to?

I think Fig. S. 1.3.14 should be included in Fig.6. 

So overall, the manuscript currently seems too descriptive. I would like to see some dicussions on the key research questions in studying the millipedes, and how a high quality genome actually contributes to elucidating those questions. 

Minor comments:

- Method section is too brief, and is almost entirely placed in Supplemental Information 1. PLoS Biology is an online journal, so space should not be a limitation. I would advice the authors to move the entire method text (S1.2) to the main manuscript, otherwise it is very difficult to follow.

- Entire Results and Discussion section is constantly referred to Supplementary information S1. Supplementary Information S1.3 mostly simply contain figures and no supplementary text, so the mention to Supplementary information S1 is not necessary, and the current mention of specific supplementary text and figures already suffice. The Supplementary Information S1 contains 35 pages, and it is very difficult to flip back and forth finding necessary information from this very broad reference. 

- chromosome assembly

 More detailed data regarding the assembly would be informative. 

 1. Genome size estimate is written to be done with GenomeScope, but without the mention of exact estimate value. What was the estimate, and does the assembly size match the estimate? If not, why?

 2. What is known about the chromosome number of the analyzed species or other species that are phylogenetically close? Looking through PubMed there seem to be a number of studies on millipede karyotyping. Does the number of pseudomolecules corresopnding to the karryotypes? The numbers 8 and 17 seem to be very close to the reported number of chromosomes. However, then, do the assembled pseudomolecules span centromeres? Do they contain telomeric repeats at the ends? These detailed information support the quality of the assembly. 

 What about the sex chromosome? Is it identifiable?

 3. Screening for contamination using Kraken is noted, but the exact results are not given. 

 4. Some comparison with previous assembly of T. corallinus would be useful. 

- Data availability

 BioProject IDs (under embargo) are given in the manuscript, but no accession for the assembly is given. Availability of assembled genome is critical for this paper.

Figure 1A: This tree does not even include chelicerates, which is mentioned in the text, so contains much less information than what is written in the text. This is totally unnecessary. 

Line 164 S. maritima species name is wrong

Line 321 Fig. probably S1.3.12, not 1.3.13

Supplementary Information S1 Italicizing species and gene names are wrong in many parts.

---

## [Decision Letter · Decision Letter 2]

3 May 2020

Dear Dr Hui,

Thank you very much for submitting a revised version of your manuscript "Millipede genomes reveal unique adaptations during myriapod evolution" for consideration as a Research Article at PLOS Biology. This revised version of your manuscript has been evaluated by the PLOS Biology editors, the Academic Editor and both of the original reviewers.

In light of the reviews (below), we offer you the opportunity to address the remaining points from the reviewers in a revised version that we anticipate should not take you very long. We will then assess your revised manuscript and your response to the reviewers' comments and we may consult the reviewers again. I should warn you that we will consult the reviewers only one more time; if they remain unsatisfied then we will not consider the manuscript further.

IMPORTANT: You'll see that while both reviewers see improvements in your previous revisions, they both find the overall structure of the paper to be unsatisfactory, and suggest ways by which you should remedy this problem. However, you'll see that reviewer #2 has some more serious criticisms, which include a) scepticism about your claimed telomeres, wanting more clarity as to how you identified them, b) further queries about your analysis of conserved synteny, which is still problematic, partly because of your choice of humans as the “other” bilaterian, and c) concerns about your functional analysis of the ozadene gland transcriptome, given previous studies. The Academic Editor has asked me to emphasise these points, and especially the comparison with other species.

We expect to receive your revised manuscript within 1 month.

**IMPORTANT - SUBMITTING YOUR REVISION**

*Resubmission Checklist*

*Published Peer Review*

*PLOS Data Policy*

*Blot and Gel Data Policy*

Sincerely,

Roli Roberts

Senior Editor

PLOS Biology

REVIEWERS' COMMENTS:

Reviewer #1:

[identifies himself as Ariel Chipman]

This is a vastly improved version of the manuscript. The authors have addressed all of my main concerns, and the manuscript is now better focused and reads much more smoothly.

The manuscript needs some very minor tweaks, after which it can be accepted without re-review.

- The transposable elements section and the genome size section are somewhat redundant. I suggest merging them into one section (starting with TEs).

- In the discussion of Ozadene glands, after giving details about the function and structure of these glands in millipedes, the authors ask what the function of the glands is. This seems like a strange question coming right after a paragraph explaining their function.

- There are still minor typos scattered throughout - myripod instead of myriapod in a few places. Furtile instead of fertile (line 381). Missing articles in a few places. I didn't go through the manuscript and identify all of these, but I suggest the authors do one more round of thorough proofreading before resubmission.

- The figures in the submitted version are all very small, and in many cases the fonts are too small even when the image is enlarged. Please follow the publishers guidelines on resolution and font size.

- In line 547-548 there is a strikethrough. Presumably the authors meant to delete this sentence.

- Supplementary figure S.1.3.3 has abbreviation for some species names but not for others in seemingly arbitrary manner. Please use full names on first mention in the table, and consistently use abbreviations after that.

- Supplementary figure S.1.3.4 - the figure legend is unclear and seems to repeat itself.

Reviewer #2:

The revised manuscript shows some improvement on restructuring and in correcting specific points addressed by myself and the other reviewer. On the other hand, I see that many points still remain in the current version.

The story is still not coherent and focused. 

MicroRNA regulation of homeobox genes should be merged with "Millipede homeobox gene and cluster" section or at least located adjascently. 

Subsection 1: "High quality genome of ..." is a descriptive section on the sequenced genomes.

Subsection 2: "Millipede homeobox gene cluster" mainly discusses the differences between centipede and millipede.

Subsection 3: "Specific duplication of argonaute protein" lacks any mention of centipedes, mainly discusses conservation within Arthropoda (but no data is given for Crustaceans and Chelicerates)

Subsection 4: "MicroRNA regulate homeobox…" mainly discusses the differences between centipede and millipede. (or at least about miR-96 and miR-283)

Subsection 5: "Conserved synteny…" discusses larger phylogenetic conservation (bilaterian)

Subsection 6: "Millipede genome size evolution" compares two millipedes

Subsection 7: "Millipede transposable elements" is mostly descriptive, with a little comparison between millipedes and centipedes (Fig.6)

Subsection 8: "Ozadene defensive gland" is mostly descriptive, with some comparision between the millipedes.

So if I were to restructure, I would first group the descriptive sections 1,7, then go on to comparison within the millipedes 6, then compare centipedes and millipedes (2,4,7), and then finally discuss larger phylogenetic-scales (3,5), 

or even better, group them according to the research question. Majority of the section still lacks biological questions. For example, there is no discussion on genome size difference. What is the possible ecological selection pressure to make the genome larger (or smaller) in one clade? How about the selection pressure to avoid TE in gene neighborhood? Why do they (if they. I will explain below) have different ozadene contents? 

* Quality of the assembly

I am still not convinced by the telomere analysis now added as Table 1.3.1e and 1.3.1f (by the way, "8 pseudomolecules" in Table 1.3.1f title should be "17 pseudomolecules"). This is very important, because this information provides the accuracy of the scaffolding process. As I have stated in my previous comments, the overall scaffold length of the presented genome is extremely good, but the contig N50 remains modest. Therefore, I am still worrying the possibility of over-scaffolding (or mis-scaffolding). Presence of centromeres and telomeres would provide some credibility on this scaffolding process.

Now, looking at tables 1.3.1e-f, the estimated number of telomeric repeats are around 100~200. The authors simply seemed to have searched for the 5-bp motif TTAGG here, but the probability of finding a 5-bp motif is 4^-5 = 1/1024. Therefore, expected number of finding a random 5bp motif within a 100kbp segment is 100,000 / 1024 = 98, or including its complement, 195, which is intriguingly in-line with the numbers presented here. So I would like to ask again: are they, and are they the only, really "repeated" motifs at the ends of chromosomes, and was the very large 100kbp segment really necessary to find these motifs? Are there centromeres and rRNA clusters in regions of "shifts" in Hi-C contact maps or around the center of the scaffolds? Can you say with confidence that these are actually telomere-to-telomere (encompassing centromeric regions which is often very very difficult to assemble and scaffold) scaffolds? Note that insect telomere is very often TTAGG, but there are also diversity in telomeric repeat motifs.

Again, I cannot test this myself since the genome sequence itself is not availle to the reviewers.

* Synteny

Previously I stated "Synteny blocks shown in Fig.2 ... is not being able to show such synteny even between Helicorthomorpha and Strigamia". This is not yet addressed in the revision. I detail my point below.

In FIg.5 Mindots=7, each colored region within each of the three circles, starting from the top right human segment counter-clockwise, decreases in length, probably corresponding to the human chromosomes. Now, left circle (Helicorthomorpha) has syntenic block coming out from chromosome 1 (brown), 12 (light green), 16 (purple), 17(greenish brown). Middle circle (Trigoniulus) has in chromosome 2 (blue-green), 12 (green), 16 (brown). Here, the only "conserved" synteny blocks are the ones in chromosome 12 and 16, and all other "synteny blocks" shared with humans are just calculation artefacts, since if the gene synteny is conserved that far into the very distant end of Chordata (i.e. humans), surely such segment MUST be conserved within the millipedes. However, when we look at the right circle (centipede Strigamia), the "synteny block" is only in chromosome 9. So NONE of the synteny blocks shared among humans and millipedes or centipedes are conserved within the myriapods. 

So again, the use of (and the only use of) humans to discuss the bilaterial conservation is not feasible. In order to discuss the "conservation", much larger number of genomes must be compared, and more phylogenetically feasible clades (such as Cephalochrdata at the root of Chordata) must be employed. At the current state, there is no statistically convincing data to suggest conservation of bilaterian synteny.

A couple of other minor points:

1. The non-consistent color coding of human chromosomes (as exemplified above) is extremely confusing

2. What happened to the Hox cluster? (it is a synteny of more than seven genes)

*Ozadene defensive gland

I find it extremely challenging for the author's notion to say T. corallinus does not have the function to produce cyanides and other chemicals in their ozdene glands. In the first genome report of T. corallinus by Kenny et al. (2015), they clearly identified many VTGl, ARSB, MRP genes within this genome. If the authors make such strong claim, these result must be carefully outlined and compared to that of Kenny et al. (2015). Moreover, it should not be that difficult of an experiment to detect (the lack of) cyanides, benzaldehydes, and quinone derivatives from T. corralinus. There are, however, reports that some millipedes do not produce these chemicals (see for example Shear et al. (2007) BIochem. Systematics & Ecology), but such claim should be backed up with more detailed analysis. 

It is not really logical to search for cyanide/benzaldehyde pathways after looking for antibacterial proteins.

"These data and analyses strongly suggest that one of the main functions of 420 millipede T. corallinus ozadene is to provide defence against pathogenic microorganisms, 421 rather than being primarily an antipredator adaptation." Really? "strongly"? even without looking for cyanide/benzaldehyde pathways??

The supplemental figure number seems to be wrong in many cases including this one. Figure S1.3.4 (noted as tyrosinase pathways) is actually hox phylogeny.

The list of abundances of peptides is not given in Supplemental Information S4, but the data still makes me worry. 

1. ATPase is just too broad. Kinases are ATPases, helicases, gyrase, some HSPs, and so on. Moreover, the top ATPase in the list (Tco_001319-T1) is a mitochondrial ATPase. It could be abundant, but abundant everywhere, not ozadene specific.

2. Top of the list tyrosinase (Tco_008220-T1) is a hemocyanin. It is also abundant, but abundant everywhere.

3. betalanin biosynthesis - should be beta-Alanine biosynthesis

4. "relative abundance of a pathway" is not a proper statistical enrichment.

Methods:

That of synteny analysis is too simple. At least some explanation of the terminology of "Mindots" is necessary.

line 595: were is italicized

---

## [Decision Letter · Decision Letter 3]

1 Jul 2020

Dear Dr Hui,

Thank you for submitting your revised Research Article entitled "Millipede genomes reveal unique adaptations during myriapod evolution" for publication in PLOS Biology. I have now obtained advice from one of the original reviewers and have discussed their comments with the Academic Editor. 

Based on the reviews, we will probably accept this manuscript for publication, assuming that you will modify the manuscript to address the remaining points raised by the reviewers. Please also make sure to address the data and other policy-related requests noted at the end of this email.

IMPORTANT:

a) Please attend to the remaining minor requests from reviewer #2.

b) Please address my Data Policy requests (further down).

We expect to receive your revised manuscript within two weeks. Your revisions should address the specific points made by each reviewer. In addition to the remaining revisions and before we will be able to formally accept your manuscript and consider it "in press", we also need to ensure that your article conforms to our guidelines. A member of our team will be in touch shortly with a set of requests. As we can't proceed until these requirements are met, your swift response will help prevent delays to publication.

*Copyediting*

*Published Peer Review History*

*Early Version*

*Submitting Your Revision*

Sincerely,

Roli Roberts

Senior Editor

PLOS Biology

DATA POLICY:

Many thanks for depositing the raw data in NCBI. However, we also require the numerical values that underlie the data summarized in the figures and results of your paper be made available in one of the following forms:

Regardless of the method selected, please ensure that you provide the individual numerical values that underlie the summary data displayed in the following figure panels as they are essential for readers to assess your analysis and to reproduce it: Figs 2, 4B, S1.3.15, S1.3.13. Please could you also supply the alignment files for all single-gene phylogenies. NOTE: the numerical data provided should include all replicates AND the way in which the plotted mean and errors were derived (it should not present only the mean/average values).

REVIEWERS' COMMENTS:

Reviewer #2:

The authors have duly addressed all of my suggestions, and the manuscript is now coherent. 

Very minor comments:

* synteny "Evidence supporting this would help to consolidate the idea that the last common ancestor of myripods retained significantly greater synteny with the last common ancestor of bilaterians than insect ancestors" This is only partially shown, because no synteny with other arthropods are shown.

* Methods: sequencing data pre-processing: optional parameters are given but no software name is provided.

* line 489 Bioanalyzer should be capitalized for its initial.

---

## [Editor Report · Decision Letter 4]

24 Aug 2020

Dear Dr Hui,

On behalf of my colleagues and the Academic Editor, Selene L Fernandez-Valverde, I am pleased to inform you that we will be delighted to publish your Research Article in PLOS Biology. 

Early Version

PRESS 

Kind regards,

Alice Musson

Publishing Editor, 

PLOS Biology

on behalf of

Roland Roberts,

Senior Editor

PLOS Biology